# Redundancy as a Structural Coordinate in Representation Learning

## Abstract

We develop a structural perspective on redundancy in learned representations, treating *redundancy* as a quantitative property of dependence organization rather than merely as inefficiency. We define redundancy as an $f$-divergence between a joint distribution and the product of its marginals, yielding a unified functional that recovers classical quantities such as mutual information and $\chi^2$-type dependence as special cases. We establish basic bounds and regularity properties of this functional, and we give a model-based endpoint argument showing that, under competing efficiency and robustness pressures, the attainable *downstream* risk profile can admit an interior optimum at a nonzero redundancy level (i.e., neither minimizing nor maximizing redundancy is optimal under the model assumptions). Empirically, we conduct controlled sweeps with masked autoencoders (MAE), organizing outcomes by a *realized* redundancy coordinate computed on frozen probe features, and we report linear-probe accuracy together with proxy-consistency checks across multiple redundancy diagnostics, including a spectral effective-rank statistic derived from covariance geometry. Together, our results—within our controlled MAE-based study—support redundancy as a measurable coordinate for analyzing representation organization in finite learning systems.

## 1 Introduction

Redundancy—statistical dependence among coordinates of a representation—is a recurrent theme in information theory and sensory coding, where it has been framed either as inefficiency to be removed or as coupling that can support reliability and robustness Shannon (1948); Barlow (2001); Narayanan et al. (2005); Tononi et al. (1994). In modern representation learning, related questions resurface in a concrete form: self-supervised and pretraining pipelines aim to produce features that are invariant to perturbations while remaining informative for downstream tasks Chen et al. (2020); He et al. (2020); Grill et al. (2020); Chen & He (2021); Caron et al. (2020); He et al. (2022). Because invariance objectives can admit trivial constant solutions, many practical methods explicitly encourage *feature diversity* by penalizing correlations, covariance, or deviations from whitening Zbontar et al. (2021); Bardes et al. (2022); Ermolov et al. (2021). These heuristics are often motivated as "redundancy reduction," yet the empirical picture is nuanced: representations can avoid outright collapse while still exhibiting substantial low-dimensional structure or heavy-tailed spectra, and downstream performance can vary non-monotonically with the degree of whitening-like behavior He & Ozay (2022); Agrawal et al. (2022); Garrido et al. (2023). This leads to a recurring practical tension: pushing aggressively toward coordinate-wise independence can alter representation geometry in ways that do not reliably track downstream risk, while leaving strong dependence unchecked can also be undesirable under other objectives He & Ozay (2022); Zbontar et al. (2021); Bardes et al. (2022). Accordingly, our paper takes a conservative stance: rather than asserting a universal "redundancy is good" or "redundancy is bad" principle, we treat redundancy as one structural axis along which representation organization and performance can be compared in controlled settings.

**Three gaps motivating this work.** Despite extensive activity around decorrelation and dependence shaping, we find three recurring sources of ambiguity in how redundancy is discussed and measured:

- **Concept conflation.** "Redundancy" is used to refer interchangeably to mutual information and its multivariate extensions, correlation/covariance structure, effective dimension (e.g., spectrum-based rank surrogates), or even neuron-level repetition—quantities that agree in special regimes but diverge in general Watanabe (1960); Doimo et al. (2022); Nanda et al. (2023); Garrido et al. (2023); Agrawal et al. (2022).

- **Knob vs. realized redundancy.** Regularization weights (a "knob" such as $\lambda_{\mathrm{red}}$ in a training objective) do not, in general, provide a reliable *coordinate* for the realized dependence of the learned representation; even within whitening-inspired SSL, the realized spectrum can change in ways that are not captured by the knob alone He & Ozay (2022); Agrawal et al. (2022); Garrido et al. (2023).

- **Endpoints dominate the discourse.** Many narratives implicitly focus on extremes (fully collapsed vs. fully decorrelated/whitened), while leaving under-developed the viewpoint of redundancy as a measurable coordinate along which one can study an attainable performance profile and ask whether intermediate levels can be preferable in stylized regimes He & Ozay (2022); Tsipras et al. (2019); Zhang et al. (2019).

**Our object: a coordinate, separated from diagnostics.** We define redundancy as an $f$-divergence from independence, $\mathcal{R}_f(U) = D_f(P_U \| \Pi_U)$ (Definition 3.1), a construction that places classical dependence measures under one umbrella and makes explicit the reference notion of independence Ali & Silvey (1966); Csiszár (1967); Watanabe (1960). Unless stated otherwise, our theory-facing coordinate is the KL special case $R = \mathcal{R}_{\mathrm{KL}}(U) = D_{\mathrm{KL}}(P_U \| \Pi_U) = \mathrm{TC}(U)$ (total correlation / multi-information) Watanabe (1960). Crucially, we separate the *coordinate* used in statements and proofs from the *practical diagnostics* used in experiments: we report a spectral proxy $\mathcal{R}_{\mathrm{spec}}$ (Definition 3.8) derived from covariance geometry as a diagnostic, and we treat agreement across diagnostics as a falsifiable validity check rather than an assumption Roy & Vetterli (2007); Garrido et al. (2023); Agrawal et al. (2022).

**Model-based theory preview.** On the theory side, we study the attainable downstream risk profile $\mathcal{E}(R)$ indexed by the redundancy coordinate $R$, and we provide stylized endpoint mechanisms giving *sufficient conditions* under which $\mathcal{E}(R)$ can admit an interior optimum at nonzero redundancy. The point is not universality, but rather to make explicit how competing pressures (e.g., efficiency- and robustness-flavored endpoints) can, in a modeled setting, be consistent with "both extremes are suboptimal."

**Experimental philosophy: protocol-first, coordinate-ordered.** Empirically, we treat redundancy control as a measurement problem: sweeps are organized by a *realized* redundancy coordinate computed on frozen probe features, not merely by a regularizer knob. We report downstream risk (linear probe) alongside reconstruction objectives (for the generative setting) and interpret discrepancies between these signals as part of the story rather than a failure of bookkeeping He et al. (2022); Garrido et al. (2023). To reduce spurious conclusions from a single proxy, we implement validity checks that test proxy-consistency across diagnostics, controllability under the sweep, and coverage of a meaningful range of realized redundancy; conclusions are explicitly conditional on these checks passing.

**Contributions.** We make the following contributions:

- **Redundancy as a structural coordinate.** We position redundancy as a quantitative descriptor of representation organization and use it to organize and interpret objective-dependent behavior within a self-supervised pipeline (generative pretraining with discriminative evaluation), as illustrated in our MAE experiments.

- **Conditional sufficient conditions for an interior optimum.** Within stylized modeled regimes, we give endpoint mechanisms that provide sufficient conditions that imply the endpoint inequalities required by our interior-optimum result, motivating an efficiency–robustness balance away from both extremes.

- **A controlled protocol in the generative regime.** We specify a masked autoencoder (MAE) protocol (Section 4) to evaluate downstream linear-probe performance alongside redundancy diagnostics across regularization sweeps with bootstrap-based uncertainty estimates.

**Roadmap.** Section 2 situates our coordinate/diagnostics separation relative to existing notions of redundancy. Section 3 develops the redundancy functional, basic properties, and the modeled mechanisms used to

motivate interior optima in attainable profiles. Section 4 presents the protocol-first empirical sweep design, diagnostics, and evaluation gates. Section 5 discusses limitations and interpretation; proofs and technical lemmas are deferred to the appendices.

## 1.1 Scope & Usage

**Scope & Usage (canonical).** Throughout, *redundancy* means statistical dependence among representation coordinates, defined as an $f$-divergence from independence: $\mathcal{R}_f(U) = D_f(P_U \| \Pi_U)$ (Definition 3.1). Unless stated otherwise, the redundancy coordinate in theorems is the KL special case $R := \mathcal{R}_{\mathrm{KL}}(U) = D_{\mathrm{KL}}(P_U \| \Pi_U) = \mathrm{TC}(U)$, applied to the representation actually used downstream (e.g., $U = \widetilde{Z}$ in Section 3.4). *Operationally*, we always take $\widetilde{Z}$ to be the output of a fixed coordinate-wise product channel $K$ applied to the learned representation $Z = \phi_\theta(X)$; this avoids differential-entropy pathologies and makes the KL/TC coordinate operational in settings where unregularized $\mathrm{TC}(Z)$ can be undefined or $+\infty$ (e.g., deterministic/continuous representations). When needed, we explicitly restrict attention to representations with $\mathcal{R}_{\mathrm{KL}}(\widetilde{Z}) < \infty$ (Definition 3.11). In this sense, our theory-facing coordinate is intended for *finite, noisy, structured* representations: finiteness is ensured by working with $\widetilde{Z}$ under a fixed product channel $K$ (e.g., quantization or independent corruption), "noisy" emphasizes that $K$ can model finite-precision or independent perturbations, and "structured" refers to nontrivial residual dependence among coordinates captured by $\mathcal{R}_{\mathrm{KL}}(\widetilde{Z}) = \mathrm{TC}(\widetilde{Z})$.

**Spectral diagnostic.** We also report $\mathcal{R}_{\mathrm{spec}}(Z)$ (Definition 3.8) as a covariance-geometry diagnostic computed from centered, per-coordinate standardized features. It is used only as a diagnostic and does not enter the proof chain for theorems stated for $\mathcal{R}_f$.

**Entropy convention for capacity-side arguments.** See Convention 1.1. Closed-form Gaussian identities for $D_{\mathrm{KL}}(P_U \| \Pi_U)$ and $D_{f_{\chi^2}}(P_U \| \Pi_U)$ are stated in the standard continuous setting and are used only when explicitly assuming joint Gaussianity.

**Experiments.** This version specifies experimental protocols and summarizes results for the E1 sweep (Section 4).

**Convention 1.1** (Entropy and quantization)**.** All entropies $H(\cdot)$ are *Shannon entropies* of discrete variables, or of fixed-resolution quantized variables $Q_\Delta(\cdot)$ at a fixed resolution $\Delta > 0$. This avoids differential-entropy pathologies and matches finite-precision representations. Here $Q_\Delta : \mathbb{R} \to \Delta\mathbb{Z}$ denotes a measurable quantization map (e.g., rounding to the nearest multiple of $\Delta$, with ties broken upward); we identify it with the induced deterministic Markov kernel $x \mapsto \delta_{Q_\Delta(x)}$. For vector-valued inputs $z \in \mathbb{R}^n$, we apply $Q_\Delta$ coordinate-wise: $(Q_\Delta(z))_i = Q_\Delta(z_i)$, yielding a product kernel $\bigotimes_{i=1}^n Q_\Delta$.

## 2 Related Works

**I. Redundancy as measurement (coordinates, estimators, diagnostics).** Dependence among representation coordinates has been quantified in many ways: via mutual information and its multivariate extensions (e.g., total correlation / multi-information), via kernel- or distance-based independence measures, and via sample-based estimators of dependence Watanabe (1960); Gretton et al. (2005); Belghazi et al. (2018). In representation learning, total-correlation-style penalties are explicitly used to encourage factorial latents in disentanglement settings Kim & Mnih (2018); Chen et al. (2018). Recent work also treats redundancy as an empirical property of trained networks, including analyses of redundant features in wide models and "diffused" redundancy in pretrained representations Doimo et al. (2022); Nanda et al. (2023); Zollikofer et al. (2024), and information-theoretic decompositions of predictive information Wollstadt et al. (2023). Our framing differs in two ways: we (i) fix a single dependence-based coordinate (an $f$-divergence from independence) for theory statements, and (ii) explicitly separate that coordinate from the diagnostics used to approximate or proxy it in practice.

**II. Redundancy reduction and decorrelation objectives in SSL (knobs).** Many successful self-supervised learning methods learn invariances while preventing collapse through architectural

or algorithmic choices (e.g., negatives, momentum targets, stop-gradient, clustering), as in Sim-CLR/MoCo/BYOL/SimSiam/SwAV Chen et al. (2020); He et al. (2020); Grill et al. (2020); Chen & He (2021); Caron et al. (2020). A complementary line of methods introduces explicit correlation/covariance shaping, ranging from cross-correlation matching (Barlow Twins) to variance–invariance–covariance regularization (VICReg) and whitening-based objectives (W-MSE) Zbontar et al. (2021); Bardes et al. (2022); Ermolov et al. (2021). Analyses of these objectives highlight that "collapse" and "whitening" are not binary states: spectra can interpolate between extremes, and eigenspectrum decay can be predictive of downstream behavior in certain regimes He & Ozay (2022); Agrawal et al. (2022). Our contribution is not a new SSL objective; rather, we treat redundancy as a *measurable coordinate* and emphasize that the regularizer weight (the knob) need not be a faithful proxy for the realized dependence level, motivating realized-coordinate ordering and validity checks.

**III. Information Bottleneck and compression viewpoints (boundary-setting).** The Information Bottleneck (IB) formalizes a trade-off between predictive information about a target and compression of the input Tishby et al. (2000); Alemi et al. (2017). A large literature applies or debates IB-inspired narratives in deep learning, including analyses based on the information plane and critiques of when compression claims do or do not appear in practice Shwartz-Ziv & Tishby (2017); Saxe et al. (2019). We use these works mainly to clarify a boundary: our redundancy coordinate measures *dependence among representation coordinates* (a divergence from independence), which is generally distinct from input compression $I(X;Z)$ or sufficiency $I(Z;Y)$. Accordingly, statements about intermediate optima in $\mathcal{E}(R)$ should be read as conditional claims about a dependence coordinate, not as claims about an "optimal compression" point.

**IV. Geometry and spectrum diagnostics (effective dimension and covariance structure).** Spectrum-based summaries of representations—including effective rank and related eigenspectrum-based criteria—are widely used as diagnostics of representation geometry and degeneracy. Effective rank provides a continuous notion of dimensionality based on spectral entropy Roy & Vetterli (2007), and recent SSL-focused work uses rank/eigenspectrum summaries as task-agnostic predictors of downstream performance or as tools for model selection Garrido et al. (2023); Agrawal et al. (2022); He & Ozay (2022). Our spectral diagnostic $\mathcal{R}_{\text{spec}}$ is in this spirit: it is a covariance-geometry proxy used for measurement and ordering, while our theorems are stated for the divergence-based redundancy coordinate. This is why our experimental protocol stresses proxy-consistency and coverage, rather than treating any single spectrum statistic as the coordinate itself.

**V. Neural Collapse (related geometry, different object).** Neural Collapse describes terminal-phase geometric regularities of supervised classifiers, including within-class variability collapse and simplex-ETF structure of class means and last-layer classifiers Papyan et al. (2020). Follow-up work studies Neural Collapse under MSE loss and its optimization landscape in simplified settings Han et al. (2022); Zhou et al. (2022). While both Neural Collapse and our diagnostics touch geometry, they operate at different levels: Neural Collapse is class-conditional and classifier-linked, whereas our redundancy coordinate is an unconditional dependence measure over coordinates of the downstream representation. We therefore position Neural Collapse as complementary context, not as an effect we claim to induce.

**VI. Theory links: robustness–capacity/efficiency trade-offs (careful positioning).** Several theory lines articulate tensions between robustness and accuracy or between different kinds of features, showing that robustness constraints can favor different representations even in stylized settings Tsipras et al. (2019); Ilyas et al. (2019); Zhang et al. (2019). At a different granularity, redundancy has also been connected to generalization behavior in wide networks and to how information is distributed across features Doimo et al. (2022); Nanda et al. (2023). Our theoretical lens is narrower: we provide sufficient endpoint mechanisms for an interior optimum in an attainable-risk profile indexed by a dependence coordinate, and we present this as one way to formalize how competing pressures can make neither extreme (fully independent vs. highly dependent) optimal under modeling assumptions.

# 3 A Redundancy Framework

**Notation.** Let $U = (U_1, \ldots, U_n)$ be a random vector taking values in $\mathcal{U} = \mathcal{X}_1 \times \cdots \times \mathcal{X}_n$. We use $n$ to denote the representation dimension throughout (e.g., $Z \in \mathbb{R}^n$ in later sections). When working in the Euclidean setting $Z \in \mathbb{R}^n$, we identify $\mathbb{R}^n \simeq \prod_{i=1}^n \mathbb{R}$ as a product measurable space and may take $\mathcal{X}_i = \mathbb{R}$ with $\mu_i$ Lebesgue measure. For each coordinate, let $\mu_i$ be a $\sigma$-finite measure on $\mathcal{X}_i$ and set $\mu := \bigotimes_{i=1}^n \mu_i$. When $P_U \ll \mu$, write its density as $p = \frac{dP_U}{d\mu}$, and for each $i$, when $P_{U_i} \ll \mu_i$, write its density as $p_i = \frac{dP_{U_i}}{d\mu_i}$. Define the product-of-marginals measure

$$\Pi_U := \bigotimes_{i=1}^n P_{U_i}, \qquad \frac{d\Pi_U}{d\mu}(u) = \prod_{i=1}^n p_i(u_i) \quad (\mu\text{-a.e.}).$$

For a positive semidefinite matrix $M$, denote by $\lambda(M)$ its eigenvalues, by $\mathrm{tr}(M)$ its trace, and by $\|\cdot\|_F$ the Frobenius norm. (Measure-theoretic details and mild regularity conditions used later are collected in Appendix C, Section C.1.)

**Eigenvalue convention.** Throughout, $\lambda(M)$ denotes the multiset of eigenvalues of $M$ (counting multiplicity); statements like $\lambda(M) \subset (a, b)$ mean every eigenvalue lies in $(a, b)$.

## Redundancy as divergence from independence

**Definition 3.1** (*$f$-divergence redundancy functional*). Let $U = (U_1, \ldots, U_n)$ be an $\mathcal{U} = \prod_{i=1}^n \mathcal{X}_i$-valued random vector with joint law $P_U$ and marginals $\{P_{U_i}\}_{i=1}^n$, and let

$$\Pi_U := \bigotimes_{i=1}^n P_{U_i}$$

denote the product-of-marginals (i.e., the law under independence).

Let $f : [0, \infty) \to \mathbb{R} \cup \{+\infty\}$ be convex with $f(1) = 0$, with the convention $f(0) := \lim_{t \downarrow 0} f(t)$ (possibly $+\infty$). We define the *redundancy* of $U$ as the $f$-divergence from independence,

$$\mathcal{R}_f(U) := D_f(P_U \| \Pi_U) := \begin{cases} \int_{\mathcal{U}} f\left(\frac{dP_U}{d\Pi_U}\right) d\Pi_U, & \text{if } P_U \ll \Pi_U, \\ +\infty, & \text{otherwise.} \end{cases} \tag{1}$$

When $P_U \ll \Pi_U$, this can be written equivalently as $\mathcal{R}_f(U) = \mathbb{E}_{\Pi_U}\left[ f\left(\frac{dP_U}{d\Pi_U}\right) \right]$. In this case $L := \frac{dP_U}{d\Pi_U} \geq 0$ $\Pi_U$-a.e. and $\mathbb{E}_{\Pi_U}[L] = 1$.

If $P_U$ and $\Pi_U$ admit densities $p$ and $\pi$ w.r.t. a common $\sigma$-finite dominating measure $\mu = \bigotimes_i \mu_i$ (i.e., $P_U \ll \mu$ and $\Pi_U \ll \mu$, which holds when each $P_{U_i} \ll \mu_i$), then the Radon-Nikodym derivative satisfies

$$\frac{dP_U}{d\Pi_U}(u) = \frac{p(u)}{\pi(u)} = \frac{p(u)}{\prod_{i=1}^n p_i(u_i)} \quad (\Pi_U\text{-a.e.}),$$

where $p_i = \frac{dP_{U_i}}{d\mu_i}$ is the marginal density and $\pi(u) = \prod_{i=1}^n p_i(u_i)$ is the density of $\Pi_U$ w.r.t. $\mu$. Assuming additionally $P_U \ll \Pi_U$ (otherwise $\mathcal{R}_f(U) = +\infty$ by equation 1), we have

$$\mathcal{R}_f(U) = \int_{\mathcal{U}} f\left(\frac{p(u)}{\prod_{i=1}^n p_i(u_i)}\right) \prod_{i=1}^n p_i(u_i)\, d\mu(u).$$

We refer to $\mathcal{R}_f$ as the *redundancy functional* and to $f$ as its *kernel*. See Appendix C, Section C.1 for further measure-theoretic details.

*Remark* 3.2 (Specializations and the role of the kernel $f$). **Role of the kernel.** The convex kernel $f$ determines how departures from independence are weighted through the likelihood ratio $L(u) := \frac{dP_U}{d\Pi_U}(u)$. We use two special cases: $f(t) = t \log t$ (total correlation / multi-information) and $f(t) = \frac{1}{2}(t-1)^2$ for $t \geq 0$ (Pearson $\chi^2$), the latter being useful for local quadratic approximations (Proposition 3.7). For the Pearson kernel $f_{\chi^2}(t) = \frac{1}{2}(t-1)^2$, one has $D_{f_{\chi^2}}(P\|Q) = \frac{1}{2}\chi^2(P\|Q)$ whenever $\chi^2(P\|Q) < \infty$.

Table 1: Notation summary.

| Symbol | Meaning |
|---|---|
| $U = (U_1, \ldots, U_n)$ | Abstract random vector in the redundancy definition; $\Pi_U = \bigotimes_i P_{U_i}$ is the product of marginals |
| $f$ | Convex $f$-divergence kernel with $f(1) = 0$ (Definition 3.1) |
| $L$ | Likelihood ratio $L = \frac{dP_U}{d\Pi_U}$ (when $P_U \ll \Pi_U$) |
| $\mathcal{R}_f(U)$ | $f$-divergence redundancy: $D_f(P_U \| \Pi_U)$ (Definition 3.1) |
| $R$ | Default redundancy coordinate (KL / total correlation); see Scope & Usage (Section 1.1). |
| $X$ | Input random variable (data). |
| $S$ | Downstream task variable/target. |
| $\theta$ | Representation parameters in $Z = \phi_\theta(X)$. |
| $g \in \mathcal{G}$ | Downstream predictor (e.g., linear probe) in a class $\mathcal{G}$. |
| $\ell_{\text{task}}$ | Task loss used to define risk in $\mathcal{E}(R)$. |
| $\varepsilon_0$ | Tolerance used in the feasible set $\Theta_{\varepsilon_0}(R)$. |
| $\Delta$ | Quantization resolution used when invoking Convention 1.1. |
| $\sigma^2$ | Noise variance in corruption models (e.g., Proposition 3.14). |
| $\log$ | Natural logarithm (nats) unless specified; $\log_2$ denotes base-2 logarithm (bits) |
| $B_0$ | Marginal entropy budget: $\sum_i H(\widetilde{Z}_i) \leq B_0$ in the capacity-side mechanism (Convention 1.1) |
| $R_{\max}$ | Upper endpoint of the target redundancy range $R \in [0, R_{\max}]$ used to index $\mathcal{E}(R)$ (chosen within a feasible interval; see Assumption 3.12). |
| $K$ | Fixed coordinate-wise product Markov channel $K = \bigotimes_{i=1}^{n} K_i$ used to define the downstream representation $\widetilde{Z}$ (Section 3.4). |
| $\widetilde{Z}$ | Downstream representation $\widetilde{Z} := K(\phi_\theta(X))$ used in the attainable profile definition (Definition 3.11). |
| $\Theta_{\varepsilon_0}(R)$ | Parameter feasible set at redundancy level $R$ (Definition 3.11); $\Gamma_{\varepsilon_0}(R)$ is the corresponding admissible pair set (Definition 3.11). |
| $\mathcal{E}(R)$ | Attainable risk profile at redundancy level $R$ (Definition 3.11) |
| $\mathcal{E}_0$ | Zero-information baseline risk (Eq. (7)) |
| $\text{TC}(U)$ | Total correlation (multi-information); in Section 3.4 the coordinate is applied to $U = \widetilde{Z}$. |
| $\mathcal{R}_{\text{KL}}(U)$ | KL redundancy: $\mathcal{R}_{\text{KL}}(U) = D_{\text{KL}}(P_U \| \Pi_U) = \text{TC}(U)$ |
| $\Sigma_Z, C$ | Covariance of centered $Z$, and its correlation matrix $C = D^{-1/2} \Sigma_Z D^{-1/2}$ with $D = \text{Diag}(\text{diag}(\Sigma_Z))$ |
| $A$ | Deviation from identity: $A := C - I$ (near-independence expansions) |
| $\mathcal{R}_{\chi^2}(Z)$ | Pearson $\chi^2$ redundancy: $\mathcal{R}_{\chi^2}(Z) = D_{f_{\chi^2}}(P_Z \| \Pi_Z)$ with $f_{\chi^2}(t) = \frac{1}{2}(t-1)^2$ |
| $\|C - I\|_F^2$ | Frobenius quadratic diagnostic for near-independence dependence geometry |
| $\mathcal{R}_{\text{spec}}(Z)$ | Spectral diagnostic: $1 - r_{\text{eff}}(Z)/n$ |
| $\lambda_{\text{red}}$ | Weight on redundancy regularizer in the training objective |

## 3.1 Basic Properties of Redundancy

**Proposition 3.3** (Nonnegativity and characterization of independence). *Let $f : [0, \infty) \to \mathbb{R} \cup \{+\infty\}$ be convex with $f(1) = 0$, with the convention $f(0) := \lim_{t \downarrow 0} f(t)$ (possibly $+\infty$). Then $\mathcal{R}_f(U) = D_f(P_U \| \Pi_U) \geq 0$. Moreover, if $f$ is strictly convex on some interval containing 1 (in particular, strict convexity in a neighborhood of 1 is sufficient), then*

$$\mathcal{R}_f(U) = 0 \iff P_U = \Pi_U,$$

*i.e., the coordinates $U_1, \ldots, U_n$ are mutually independent.*

*Proof.* See Appendix C. □

**Proposition 3.4** (Data processing inequality (DPI)). *Let $K_i$ be Markov kernels from $\mathcal{X}_i$ to measurable spaces $(\mathcal{Y}_i, \mathcal{B}_i)$ (with their Borel or given $\sigma$-algebras), and let $K := \bigotimes_{i=1}^{n} K_i$ be the product kernel on $\mathcal{U} \to \prod_{i=1}^{n} \mathcal{Y}_i$. Let $Y$ denote the random vector obtained by passing $U$ through $K$, i.e., $P_Y = P_U K$. Then $\Pi_Y = \Pi_U K$ (since $K = \bigotimes_{i=1}^{n} K_i$ acts independently on coordinates) and*

$$\mathcal{R}_f(Y) = D_f(P_Y \| \Pi_Y) \leq D_f(P_U \| \Pi_U) = \mathcal{R}_f(U).$$

*In particular, if $\mathcal{R}_f(U) = +\infty$ the inequality holds trivially in the extended reals; the nontrivial case is when $\mathcal{R}_f(U) < \infty$.*

*Proof.* See Appendix C. □

**Proposition 3.5** (Elementary upper bound under bounded likelihood ratio). *Assume $P_U \ll \Pi_U$ and let $L = \frac{dP_U}{d\Pi_U}$. If $m \leq L \leq M$ $\Pi_U$-a.e. for some $0 \leq m \leq 1 \leq M$, then $\mathcal{R}_f(U) = \mathbb{E}_{\Pi_U}[f(L)] \leq \sup_{t \in [m,M]} f(t)$ (finite if $f$ is finite on $[m, M]$). In particular, for $f(t) = t \log t$, since $\mathbb{E}_{\Pi_U}[L] = 1$ implies $m \leq 1 \leq M$, we have $\mathcal{R}_{\mathrm{KL}}(U) \leq M \log M$. Moreover, since $L \leq M$ implies $L \log L \leq L \log M$ and $\mathbb{E}_{\Pi_U}[L] = 1$, we also have the tighter bound $\mathcal{R}_{\mathrm{KL}}(U) = \mathbb{E}_{\Pi_U}[L \log L] \leq \log M$. Remark. This bounded-likelihood-ratio condition is a strong sanity assumption and is not expected to literally hold for high-dimensional learned representations; it is used only to illustrate basic boundedness behavior.*

*Proof.* See Appendix C. □

### 3.2 Gaussian and Quadratic Approximations

**Proposition 3.6** (Gaussian total correlation). *Let $U \sim \mathcal{N}(0, \Sigma)$ with $\Sigma \succ 0$ and let $C = \mathrm{corr}(U)$ be its correlation matrix (so $C \succ 0$, i.e., symmetric positive definite). (In particular, this identity depends on the correlation matrix $C$, not the raw covariance $\Sigma$.) Then the redundancy under the Kullback–Leibler kernel equals the Gaussian total correlation:*

$$\mathcal{R}_{\mathrm{KL}}(U) = D_{\mathrm{KL}}(P_U \| \Pi_U) = -\tfrac{1}{2} \log \det C.$$

*Proof.* See Appendix C. □

**Proposition 3.7** (Local quadratic approximation under weak dependence). *Let $U \sim \mathcal{N}(0, \Sigma)$ be centered Gaussian with correlation matrix $C = \mathrm{corr}(U)$, and write $A := C - I$. Assume $\|A\|_2 < 1$ (equivalently, $\lambda(C) \subset (0, 2)$, since $C$ is symmetric). This condition ensures the series expansion converges; it is used here only to motivate the local quadratic form and is not empirically verified for learned representations (which need not satisfy weak dependence).*

*(i) KL (total-correlation) redundancy.*

$$\mathcal{R}_{\mathrm{KL}}(U) = -\tfrac{1}{2} \log \det C.$$

*Moreover, as $\|C - I\|_F \to 0$,*

$$\mathcal{R}_{\mathrm{KL}}(U) = \tfrac{1}{4} \|C - I\|_F^2 + O(\|C - I\|_F^3). \tag{2}$$

*In particular, since $C$ is a correlation matrix, $\mathrm{diag}(A) = 0$ and $\|C - I\|_F^2 = \|A\|_F^2 = \sum_{i \neq j} C_{ij}^2$.*

*(ii) Quadratic ($\chi^2$) redundancy. For the quadratic kernel $f_{\chi^2}(t) = \frac{1}{2}(t-1)^2$, note that $\chi^2$ (hence $\mathcal{R}_{\chi^2}$) can be infinite outside this local weak-dependence regime; here the assumption $\|A\|_2 < 1$ is used to ensure finiteness and justify the expansion.*

$$\mathcal{R}_{\chi^2}(U) = D_{f_{\chi^2}}(P_U \| \Pi_U) = \tfrac{1}{4} \|C - I\|_F^2 + O(\|C - I\|_F^3), \tag{3}$$

*as $\|C - I\|_F \to 0$.*

*Consequently, in the jointly Gaussian weak-dependence regime, $\mathcal{R}_{\mathrm{KL}}$ and $\mathcal{R}_{\chi^2}$ agree to second order:*

$$\mathcal{R}_{\mathrm{KL}}(U) = \mathcal{R}_{\chi^2}(U) + O(\|C - I\|_F^3),$$

*where the implicit constants in $O(\cdot)$ depend on $n$ and a local bound on $\|C - I\|_2$.*

*Proof.* See Appendix C. □

### 3.3 Spectral redundancy as a geometric diagnostic

**Definition 3.8** (Spectral redundancy). Let $Z \in \mathbb{R}^n$ be a random vector with covariance matrix $\Sigma_Z = \mathrm{Cov}(Z) \succeq 0$ (computed on centered features), and define the diagonal variance matrix $D := \mathrm{Diag}(\mathrm{diag}(\Sigma_Z))$. Assume $\mathrm{Var}(Z_i) > 0$ for all coordinates so that $D$ is invertible. (If some $\mathrm{Var}(Z_i) = 0$, we drop the zero-variance coordinates and redefine $n$ accordingly; after this reduction, $\Sigma_Z \succ 0$.) Define the correlation matrix $C$ by

$$C := D^{-1/2}\Sigma_Z D^{-1/2}.$$

Let $\lambda_1, \ldots, \lambda_n \geq 0$ be the eigenvalues of $C$ (so $\sum_{j=1}^n \lambda_j = \mathrm{tr}(C) = n$). Define the normalized spectrum

$$\tilde{\lambda}_i = \frac{\lambda_i}{\sum_{j=1}^n \lambda_j}, \qquad \sum_{i=1}^n \tilde{\lambda}_i = 1,$$

and the associated spectral entropy and effective rank

$$H_\lambda(Z) = -\sum_{i=1}^n \tilde{\lambda}_i \log \tilde{\lambda}_i, \qquad r_{\mathrm{eff}}(Z) = \exp\big(H_\lambda(Z)\big).$$

We use the convention $0 \cdot \log 0 := 0$. We define the *spectral redundancy diagnostic* as the normalized deficit in effective rank,

$$\mathcal{R}_{\mathrm{spec}}(Z) = 1 - \frac{r_{\mathrm{eff}}(Z)}{n}, \qquad \mathcal{R}_{\mathrm{spec}}(Z) \in \left[0,\ 1 - \frac{1}{n}\right].$$

**Standardization.** In our protocols we compute $C$ from centered, per-coordinate standardized features.

**Relationship to covariance geometry.** In jointly Gaussian weak-dependence regimes, both $\mathcal{R}_{\mathrm{KL}}$ and $\mathcal{R}_{\chi^2}$ admit second-order covariance-geometry expansions in $\|C - I\|_F^2$ (Proposition 3.7), which motivates reporting spectrum concentration diagnostics such as $\mathcal{R}_{\mathrm{spec}}$.

**Proposition 3.9** (Bounds and extremal cases). *The diagnostic $\mathcal{R}_{\mathrm{spec}}(Z)$ satisfies*

$$\mathcal{R}_{\mathrm{spec}}(Z) = 0 \quad \Longleftrightarrow \quad \tilde{\lambda}_1 = \cdots = \tilde{\lambda}_n = \frac{1}{n},$$

*i.e., the normalized spectrum is uniform (equivalently, $r_{\mathrm{eff}}(Z) = n$). Moreover,*

$$\mathcal{R}_{\mathrm{spec}}(Z) = 1 - \frac{1}{n} \quad \Longleftrightarrow \quad \tilde{\lambda}_1 = 1,\ \tilde{\lambda}_{i>1} = 0,$$

*i.e., the spectrum is fully concentrated on one eigenvalue (equivalently, $\mathrm{rank}(\Sigma_Z) = 1$), in which case $r_{\mathrm{eff}}(Z) = 1$.*

*Proof.* Since $\tilde{\lambda}$ lies in the probability simplex, the Shannon entropy $H_\lambda(Z) = -\sum_i \tilde{\lambda}_i \log \tilde{\lambda}_i$ satisfies $0 \leq H_\lambda(Z) \leq \log n$, with equality $H_\lambda(Z) = \log n$ if and only if the spectrum is uniform and $H_\lambda(Z) = 0$ if and only if it is concentrated on one coordinate. Exponentiating yields $r_{\mathrm{eff}}(Z) = \exp(H_\lambda(Z)) \in [1, n]$ with the same equality cases. Substituting into $\mathcal{R}_{\mathrm{spec}}(Z) = 1 - r_{\mathrm{eff}}(Z)/n$ gives the stated bounds and extremizers. □

### 3.4 Redundancy balance and the existence of an interior optimum

Let $Z = \phi_\theta(X)$ be a learned representation (where $\phi_\theta$ denotes the encoder, distinct from the divergence kernel $f$) and let $S$ denote the downstream task variable. To avoid ambiguity about which representation is *used* downstream, and to make the KL/TC redundancy coordinate operational in settings where $\mathrm{TC}(Z)$ may be undefined or $+\infty$ (e.g., deterministic/continuous $Z$), we allow a fixed coordinate-wise product Markov channel (cf. Proposition 3.4) to be applied before prediction. Assume each coordinate $Z_i$ takes values in a measurable space $\mathcal{X}_i$ and each $K_i$ is a Markov kernel from $\mathcal{X}_i$ to a measurable space $\mathcal{Y}_i$, so $K = \bigotimes_{i=1}^n K_i$ is a well-defined product kernel. Define the downstream representation

$$\widetilde{Z} \;:=\; K(Z) = K(\phi_\theta(X)).$$

(This $K$ is a fixed instance of the generic product kernel in Proposition 3.4.) In this section the redundancy coordinate is computed on the downstream representation:

$$R \;:=\; \mathcal{R}_{\mathrm{KL}}(\widetilde{Z}) = D_{\mathrm{KL}}(P_{\widetilde{Z}}\|\Pi_{\widetilde{Z}}) = \mathrm{TC}(\widetilde{Z})$$

(Definition 3.1). Other quantities such as $\mathcal{R}_{\chi^2}$ or $\mathcal{R}_{\mathrm{spec}}$ are used only as computational diagnostics in experiments. Throughout this subsection, write $\mathcal{R}(\cdot) := \mathcal{R}_{\mathrm{KL}}(\cdot)$ for the redundancy coordinate.

*Remark* 3.10 (Coordinate clarification (TC vs. diagnostics)). All formal statements in this subsection are in the KL/TC coordinate $R = \mathrm{TC}(\widetilde{Z})$. Our training protocol regularizes the diagnostic $\mathcal{R}_{\mathrm{spec}}$; an interior optimum in $\mathcal{R}_{\mathrm{spec}}$ does not, by itself, imply an interior optimum in $R$ without additional proxy-consistency conditions.

Fix an operational tolerance $\varepsilon_0 > 0$ and choose a redundancy range endpoint $R_{\max} > 0$ within a redundancy interval for which the constraint sets are feasible (within tolerance); the interval $[0, R_{\max}]$ is treated as the target range of redundancy levels under consideration.

**Notation convention.** For brevity, when $\theta$ induces a distribution $P_{\widetilde{Z}}$ on the downstream representation $\widetilde{Z} = K(\phi_\theta(X))$, we write $\mathcal{R}(K(\phi_\theta(X)))$ to mean $\mathcal{R}_{\mathrm{KL}}(P_{\widetilde{Z}})$, the redundancy of the distribution (not of a particular realization). This is a deterministic function of $\theta$ (via the induced distribution).

**Definition 3.11** (Attainable risk profile). **Feasible set (fixed tolerance).** For $R \in [0, R_{\max}]$ (the target redundancy level), define

$$\Theta_{\varepsilon_0}(R) \;:=\; \left\{\theta :\; \big|\mathcal{R}(K(\phi_\theta(X))) - R\big| \le \varepsilon_0\right\},$$

and set

$$\Gamma_{\varepsilon_0}(R) \;:=\; \left\{(\theta, g) :\; \theta \in \Theta_{\varepsilon_0}(R),\; g \in \mathcal{G},\; \mathcal{R}(K(\phi_\theta(X))) < \infty,\; \mathbb{E}\big[\big|\ell_{\mathrm{task}}(S, g(K(\phi_\theta(X)))) \big|\big] < \infty\right\}.$$

Assume $\ell_{\mathrm{task}}$ is measurable and that for admissible $(\theta, g)$, the composition $(S, X) \mapsto \ell_{\mathrm{task}}(S, g(K(\phi_\theta(X))))$ is measurable so the expectation is well-defined.

**Attainable risk.** Define the attainable risk at redundancy level $R$ by

$$\mathcal{E}(R) \;:=\; \inf_{(\theta, g) \in \Gamma_{\varepsilon_0}(R)} \mathbb{E}[\ell_{\mathrm{task}}(S, g(K(\phi_\theta(X))))], \qquad R \in [0, R_{\max}],$$

with the convention that $\mathcal{E}(R) = +\infty$ if $\Gamma_{\varepsilon_0}(R)$ is empty, where $\mathcal{G}$ denotes the downstream predictor class (e.g., linear probes in our experiments).

**Practice-facing interpretation (achievable profile).** Definition 3.11 is an idealized value function. In practice, one typically obtains a *family* of trained solutions indexed by a penalty or constraint (e.g., $\lambda$ in a Lagrangian), and thus observes a set of achievable pairs $(R, \mathrm{risk})$ under a fixed training pipeline. The profile $\mathcal{E}(R)$ should be read as the lower envelope of such achievable pairs at tolerance $\varepsilon_0$, not as a literal claim that deep-net optimization ranges over compact parameter sets. **Note (endpoint $R = 0$ under tolerance).** Since $\varepsilon_0 > 0$ is fixed, the feasible set $\Gamma_{\varepsilon_0}(0)$ consists of representations with $|\mathcal{R}(K(\phi_\theta(X)))| \le \varepsilon_0$; thus $\mathcal{E}(0)$ is well-defined as an optimization problem even if exact redundancy 0 is not attainable (and equals $+\infty$ if $\Gamma_{\varepsilon_0}(0) = \emptyset$ by convention).

**Assumption 3.12** (Well-posedness and attainment of $\mathcal{E}$). **(a) Well-posedness.** For each $R \in [0, R_{\max}]$, $\Gamma_{\varepsilon_0}(R) \neq \emptyset$ and $\mathcal{E}(R) < \infty$. **(b) Attainment of minimum over $R$.** The profile $\mathcal{E} : [0, R_{\max}] \to \mathbb{R}$ (as a function of the redundancy coordinate $R$) attains at least one global minimum on $[0, R_{\max}]$, i.e., there exists $R^* \in [0, R_{\max}]$ such that $\mathcal{E}(R^*) = \inf_{R \in [0, R_{\max}]} \mathcal{E}(R)$. **(c) Lower semicontinuity.** The profile $\mathcal{E} : [0, R_{\max}] \to \mathbb{R}$ is lower semicontinuous. Standard sufficient conditions (compactness/continuity, including continuity of $\theta \mapsto \mathcal{R}(K(\phi_\theta(X)))$ and a dominated envelope for the task loss) are given in Appendix A. This is a modeling/well-posedness layer ensuring $\mathcal{E}$ is a proper attained profile; it is not meant as a literal description of deep-net parameter spaces.

*Remark* 3.13 (Operational view). Optimizing the training loss plus $\lambda \mathcal{R}$ can be interpreted as a Lagrangian relaxation that selects a point on the downstream profile $\mathcal{E}(R)$.

**Theory vs. practice.**   Operationally, one may implement the redundancy coordinate using a fixed tolerance $\varepsilon_0$ (as above) or via a Lagrangian penalty; the mathematical statements treat $R$ as an order parameter at the resolution of interest and do not require an $\inf_{\varepsilon \downarrow 0}$ limit inside the definition.

**Operational checklist (practice-facing proxies).**   In a discrete sweep with a fixed pipeline, one may approximate the objects in Definition 3.11 by:

- **Lower envelope:** treat $\mathcal{E}(R)$ as the empirical lower envelope of achieved $(\widehat{R}, \widehat{\text{risk}})$ pairs binned along a realized redundancy axis.

- **Feasibility sets:** treat $\Gamma_{\varepsilon_0}(R)$ as the subset of runs whose realized redundancy lies within a tolerance band around $R$ (or within a fixed-width bin).

- **Attainment proxy:** treat "attainment" as "there exists a stable best regime" after filtering out divergent/unstable runs (e.g., via loss-slope, finite-metric checks, and seed-based uncertainty).

**Proposition 3.14** (Strict improvability of the low-redundancy endpoint under component-wise independent corruption). *Fix squared-error loss $\ell(s, \hat{s}) = \|s - \hat{s}\|_2^2$ and consider the following linear-Gaussian model. Let $S \in \mathbb{R}^d$ be centered Gaussian with diagonal covariance $\mathrm{Cov}(S) = \mathrm{diag}(\nu_1, \ldots, \nu_d)$ and $\nu_1 > 0$. Assume $n \geq d + 1$ (so that the representation has at least one coordinate beyond those encoding $S$, which is needed to accommodate a redundant copy in the construction below). Let the representation be*

$$Z = WS, \qquad W \in \mathbb{R}^{n \times d},$$

*followed by component-wise independent corruption*

$$\widetilde{Z} = Z + \xi, \qquad \xi \sim \mathcal{N}(0, \sigma^2 I_n), \ \sigma^2 > 0,$$

*and restrict the decoder to be linear: $\hat{S} = B\widetilde{Z}$. This proposition corresponds to the special case where the downstream channel $K$ is additive independent Gaussian corruption (a coordinate-wise product kernel).*

***Per-coordinate gain constraint.*** *Assume each row $w_i^\top$ of $W$ satisfies $\|w_i\|_2 \leq 1$.*

***Redundancy coordinate (KL / total correlation).*** *We take the redundancy coordinate to be the Kullback–Leibler special case of Definition 3.1:*

$$R := \mathcal{R}_{\mathrm{KL}}(\widetilde{Z}) = D_{\mathrm{KL}}(P_{\widetilde{Z}} \,\|\, \Pi_{\widetilde{Z}}) = \mathrm{TC}(\widetilde{Z}).$$

*Define the (toy-model) optimal risk at redundancy level $R$ by*

$$\mathcal{E}_{\mathrm{toy}}(R) := \inf_{(W, B): \, \mathcal{R}_{\mathrm{KL}}(\widetilde{Z}) = R} \mathbb{E}\big[\|S - B\widetilde{Z}\|_2^2\big],$$

*where the infimum is taken over $(W, B)$ satisfying the gain constraint. This toy profile uses an exact redundancy constraint and is used only to witness strict improvability near $R = 0$ as an endpoint mechanism.*

*Then for any fixed $\sigma^2 > 0$, exact independence in this KL redundancy coordinate is strictly improvable: there exists a family $\{\widetilde{Z}_\alpha\}_{\alpha>0}$ with $\mathcal{R}_{\mathrm{KL}}(\widetilde{Z}_\alpha) \downarrow 0$ as $\alpha \downarrow 0$ such that, for every $\alpha > 0$,*

$$\inf_B \; \mathbb{E}\big[\|S - B\widetilde{Z}_\alpha\|_2^2\big] \; < \; \inf_B \; \mathbb{E}\big[\|S - B\widetilde{Z}_0\|_2^2\big],$$

*where $\mathcal{R}_{\mathrm{KL}}(\widetilde{Z}_0) = 0$. Consequently, whenever the constraint set is nonempty in a neighborhood of 0, $R = 0$ cannot be a local minimizer of $\mathcal{E}_{\mathrm{toy}}(R)$ in this model.*

*Proof.* Assume $n \geq d + 1$ and define, for $\alpha \in (0, 1]$,

$$Z_0 = (S_1, \ldots, S_d, 0, \ldots, 0) \in \mathbb{R}^n, \qquad Z_\alpha = (S_1, \ldots, S_d, \alpha S_1, 0, \ldots, 0) \in \mathbb{R}^n.$$

Both are realizable under the gain constraint by taking rows of $W$ equal to $e_1^\top, \ldots, e_d^\top, \alpha e_1^\top, 0, \ldots, 0$.

Since $\mathrm{Cov}(S)$ is diagonal and $S$ is Gaussian, the coordinates of $Z_0$ are independent. Therefore the corrupted vector $\widetilde{Z}_0 = Z_0 + \xi$ has independent coordinates as well, hence $\mathcal{R}_{\mathrm{KL}}(\widetilde{Z}_0) = 0$. Indeed, the pairs $(Z_{0,i}, \xi_i)$ are independent across $i$ since $S$ has independent coordinates and $\xi$ is independent with independent components.

For $\widetilde{Z}_\alpha = Z_\alpha + \xi$, the only dependence is between the two coordinates

$$Y_1 = S_1 + \xi_1, \qquad Y_2 = \alpha S_1 + \xi_{d+1},$$

which are jointly Gaussian with correlation

$$\rho(\alpha) = \frac{\mathrm{Cov}(Y_1, Y_2)}{\sqrt{\mathrm{Var}(Y_1)\mathrm{Var}(Y_2)}} = \frac{\alpha \nu_1}{\sqrt{(\nu_1 + \sigma^2)(\alpha^2 \nu_1 + \sigma^2)}}.$$

By Proposition 3.6, the total correlation of this bivariate Gaussian equals

$$\mathcal{R}_{\mathrm{KL}}(\widetilde{Z}_\alpha) = \mathrm{TC}(\widetilde{Z}_\alpha) = \mathrm{TC}(Y_1, Y_2) = -\tfrac{1}{2}\log\big(1 - \rho(\alpha)^2\big) \xrightarrow[\alpha \downarrow 0]{} 0,$$

since $\rho(\alpha) \to 0$ as $\alpha \downarrow 0$.

Under joint Gaussianity and squared loss, the optimal linear decoder coincides with the MMSE (LMMSE) estimator. The only affected coordinate is $S_1$, for which the decoder effectively observes $(Y_1, Y_2)$ defined above with independent noise variance $\sigma^2$. The resulting posterior variance (standard LMMSE result) is

$$\mathrm{Var}(S_1 \mid Y_1, Y_2) = \left(\nu_1^{-1} + \tfrac{1+\alpha^2}{\sigma^2}\right)^{-1},$$

which is strictly smaller than the baseline $\left(\nu_1^{-1} + \tfrac{1}{\sigma^2}\right)^{-1}$ for every $\alpha > 0$. All other coordinates of $S$ are unchanged, hence the overall optimal risk is strictly smaller for $Z_\alpha$ than for $Z_0$ while $\mathcal{R}_{\mathrm{KL}}(\widetilde{Z}_\alpha) \downarrow 0$. A complete derivation is given in Appendix B. $\square$

*Remark* 3.15 (Bridge to tolerance-based framework). Proposition 3.14 uses an exact redundancy constraint in the toy profile $\mathcal{E}_{\mathrm{toy}}$. To connect this to the tolerance-based Definition 3.11: if $\mathcal{E}_{\mathrm{toy}}$ is continuous and strictly improvable at $R = 0$ (i.e., $\mathcal{E}_{\mathrm{toy}}(R') < \mathcal{E}_{\mathrm{toy}}(0)$ for some $R' > 0$), then for sufficiently small $\varepsilon_0 > 0$, the tolerance-based profile can satisfy $\mathcal{E}(R) < \mathcal{E}(0)$ for $R$ near $R'$. This relies on the regularity layer ensuring lower semicontinuity of the profile $R \mapsto \mathcal{E}(R)$ (Appendix A / Assumption 3.12).

*Continuity verification for the linear-Gaussian model.* In the setting of Proposition 3.14, continuity of $\mathcal{E}_{\mathrm{toy}}(R)$ in $R$ follows from standard properties of the LMMSE estimator: both the optimal risk (posterior variance) and the redundancy coordinate $\mathcal{R}_{\mathrm{KL}}(\widetilde{Z}) = -\tfrac{1}{2}\log\det C$ are continuous functions of the correlation matrix $C$, which in turn depends continuously on the encoding matrix $W$ under the linear-Gaussian model. Hence the bridge applies to Proposition 3.14.

*Remark* 3.16 (Corruption vs. profile). Proposition 3.14 concerns a corruption model $\widetilde{Z} = Z + \xi$ with independent noise. It fits the profile definition above by taking the fixed channel $K$ to be additive independent Gaussian corruption and by taking $\mathcal{G}$ to be the class of linear decoders.

Proposition 3.14 provides a concrete stylized mechanism in which the low-redundancy endpoint is strictly improvable in a KL/TC coordinate computed on the downstream representation $\widetilde{Z}$.

The capacity-side Lemma 3.23 (Section 3.7) provides one concrete sufficient mechanism for the high-redundancy endpoint condition in Lemma 3.17 under the quantized-entropy convention of Convention 1.1.

Under Assumption 3.12, if the endpoint conditions in Lemma 3.17 hold (which can be witnessed in specific stylized settings via mechanisms such as Proposition 3.14 and Lemma 3.23), then at least one global minimizer lies in the interior. The following lemma records this structural implication.

**Lemma 3.17** (Interior minimizer under endpoint strictness). *Assume Assumption 3.12.*

*Assume:*

1. ***(Low-redundancy endpoint strictness).*** *There exists $\delta_L \in (\varepsilon_0, R_{\max}]$ (requiring $\varepsilon_0 < R_{\max}$) such that*

$$\inf_{R \in (\varepsilon_0, \delta_L]} \mathcal{E}(R) < \mathcal{E}(0). \tag{4}$$

2. ***(High-redundancy endpoint is non-optimal).*** *Let $\mathcal{E}_0$ be the zero-information baseline risk defined in equation 7, and assume the task is nontrivial in the sense that*

$$\inf_{R \in [0, R_{\max})} \mathcal{E}(R) < \mathcal{E}_0 \qquad and \qquad \mathcal{E}(R_{\max}) \geq \mathcal{E}_0. \tag{5}$$

*Then $\mathcal{E}$ attains at least one global minimum at an interior point $R^* \in (0, R_{\max})$.*

**Clarification.** Here $\mathcal{E}(0)$ refers to the tolerance-feasible near-zero redundancy set $\Gamma_{\varepsilon_0}(0)$ in Definition 3.11. Thus condition (1) compares the "near-zero" regime $|\mathcal{R}| \leq \varepsilon_0$ against the "slightly positive" regime $R \in (\varepsilon_0, \delta_L]$; the condition is meaningful because it asserts that moving beyond the tolerance band around zero strictly improves risk.

*Remark* 3.18 (Interpretation). Lemma 3.17 asserts existence of at least one interior minimizer; it does not address uniqueness.

*Proof.* By Assumption 3.12, $\mathcal{E}$ attains a global minimum at some $\widehat{R} \in [0, R_{\max}]$.

If $\widehat{R} = 0$, then $\mathcal{E}(0) \leq \mathcal{E}(R)$ for all $R$, which contradicts equation 4. Hence $\widehat{R} \neq 0$.

If $\widehat{R} = R_{\max}$, then $\mathcal{E}(R_{\max}) \leq \mathcal{E}(R)$ for all $R$. But equation 5 implies $\mathcal{E}(R_{\max}) \geq \mathcal{E}_0$ while $\inf_{R \in [0, R_{\max})} \mathcal{E}(R) < \mathcal{E}_0$, a contradiction. Hence $\widehat{R} \neq R_{\max}$.

Therefore $\widehat{R} \in (0, R_{\max})$; set $R^* := \widehat{R}$. □

*Remark* 3.19 (Local strictness). If $\mathcal{E}$ is twice differentiable at an interior minimizer $R^*$ and $\mathcal{E}''(R^*) > 0$, then $R^*$ is a strict local minimizer by the usual second-order sufficient condition.

## 3.5 Connections to classical measures and practical proxies

**Total correlation (multi-information).** For the Kullback–Leibler kernel $f(t) = t \log t$, Definition 3.1 yields

$$\mathcal{R}_{\mathrm{KL}}(U) = D_{\mathrm{KL}}(P_U \| \Pi_U),$$

i.e., the standard *total correlation* (multi-information). When the (entropic) quantities are well-defined and finite (e.g., discrete $U$),

$$\mathcal{R}_{\mathrm{KL}}(U) = \sum_{i=1}^{n} H(U_i) - H(U).$$

Moreover, $\mathcal{R}_{\mathrm{KL}}(U) = 0$ if and only if $U_1, \ldots, U_n$ are mutually independent (Proposition 3.3).

**Quadratic (covariance/correlation) proxy.** For the Pearson kernel $f(t) = \frac{1}{2}(t-1)^2$, the redundancy becomes the Pearson $\chi^2$-divergence $\mathcal{R}_{\chi^2}(U) = D_{f_{\chi^2}}(P_U \| \Pi_U)$. In a weak-dependence regime where a second-order expansion around independence is meaningful, one obtains a local quadratic approximation. In particular, for the Gaussian instantiation in Proposition 3.7,

$$\mathcal{R}_{\chi^2}(U) = \tfrac{1}{4} \|C - I\|_F^2 + o\big(\|C - I\|_F^2\big),$$

where $C = \mathrm{corr}(U)$ after standardizing to unit variances. We use such quadratic forms as differentiable geometry-inspired surrogates; outside local regimes they should not be interpreted as exact divergences. *Locality reminder.* Unlike TC, the $\chi^2$ divergence (hence $\mathcal{R}_{\chi^2}$) can be infinite outside weak-dependence regimes; we use it only as a local proxy when it is finite and numerically stable.

**Spectral diagnostic.** We also report $\mathcal{R}_{\mathrm{spec}}(Z)$ (Definition 3.8) as a computational diagnostic of spectrum concentration.

*Remark* 3.20 (What $\mathcal{R}_{\mathrm{spec}}$ does (and does not) capture). $\mathcal{R}_{\mathrm{spec}}$ is sensitive to *effective rank* (spectrum concentration) of the correlation matrix, whereas $\|C-I\|_F^2 = \sum_{i \neq j} C_{ij}^2$ measures total off-diagonal correlation energy. They are not equivalent and can disagree (e.g., many small correlations can yield large $\|C - I\|_F^2$ with near-uniform spectrum, while a low-rank correlation structure can yield large $\mathcal{R}_{\mathrm{spec}}$ without uniformly large pairwise correlations).

## 3.6 Practical lemmas for boundedness during training

**Lemma 3.21** (Spectral boundedness under norm control). *Assume in addition that all marginal variances are positive so that the correlation matrix in Definition 3.8 is well-defined. Let $Z = WX \in \mathbb{R}^n$ with $\|W\|_2 \leq M$ and $\mathbb{E}\|X\|_2^2 \leq \sigma^2$. Then the covariance $\Sigma_Z$ satisfies*

$$\mathrm{tr}(\Sigma_Z) \leq M^2 \sigma^2.$$

*Consequently, the normalized eigenvalue spectrum is well defined and $r_{\mathrm{eff}}(Z) \in [1, n]$, hence*

$$0 \leq \mathcal{R}_{\mathrm{spec}}(Z) \leq 1 - \tfrac{1}{n}.$$

*Proof.* Since $\Sigma_Z = W \Sigma_X W^\top$ and $\Sigma_X \succeq 0$,

$$\mathrm{tr}(\Sigma_Z) = \mathrm{tr}(W^\top W \Sigma_X) \leq \|W^\top W\|_2 \, \mathrm{tr}(\Sigma_X) = \|W\|_2^2 \, \mathbb{E}\|X - \mathbb{E}X\|_2^2 \leq \|W\|_2^2 \, \mathbb{E}\|X\|_2^2 \leq M^2 \sigma^2.$$

The bounds on $\mathcal{R}_{\mathrm{spec}}$ then follow from $r_{\mathrm{eff}}(Z) = \exp(H_\lambda(Z)) \in [1, n]$. $\qquad\square$

**Lemma 3.22** (Local KL–Frobenius lower bound (non-asymptotic)). *Let $U \sim \mathcal{N}(0, \Sigma)$ be a centered Gaussian random vector with correlation matrix $C = \mathrm{corr}(U)$, and write $C = I + A$ with $A = C - I$. Assume $\|A\|_2 \leq \rho$ for some fixed $\rho \in (0, 1)$. Then there exists $\varepsilon > 0$ (depending only on $n$ and $\rho$) such that whenever $\|C - I\|_F \leq \varepsilon$,*

$$\mathcal{R}_{\mathrm{KL}}(U) = -\tfrac{1}{2} \log \det C \geq \tfrac{1}{8} \|C - I\|_F^2.$$

*An explicit sufficient condition is $\varepsilon \leq \min\{1, 3(1 - \rho)/(4\sqrt{n})\}$. This identity is Gaussian-specific; for non-Gaussian distributions, total correlation (and KL-to-product-marginals) is not determined solely by the correlation matrix.*

*Proof.* With $\|A\|_2 \leq \rho < 1$, the expansion $\log(I + A) = \sum_{k \geq 1} \frac{(-1)^{k+1}}{k} A^k$ implies

$$-\log \det(I + A) = -\mathrm{tr} \log(I + A) = \sum_{k \geq 1} \frac{(-1)^k}{k} \mathrm{tr}(A^k).$$

Since $C$ is a correlation matrix, $\mathrm{tr}(A) = 0$, and $\mathrm{tr}(A^2) = \|A\|_F^2$. The remainder $\sum_{k \geq 3} \frac{(-1)^k}{k} \mathrm{tr}(A^k)$ is $O(\|A\|_F^3)$ as $\|A\|_F \to 0$ under the standing assumption $\|A\|_2 \leq \rho < 1$. Indeed, for $k \geq 3$ we may bound

$$|\mathrm{tr}(A^k)| \leq \|A^k\|_* \leq \sqrt{n} \|A^k\|_F \leq \sqrt{n} \|A\|_2^{k-2} \|A\|_F^2,$$

where $\|\cdot\|_*$ denotes the nuclear (trace) norm and we used $|\mathrm{tr}(B)| \leq \|B\|_*$, $\|B\|_* \leq \sqrt{n}\|B\|_F$, and $\|A^k\|_F = \|A^{k-2}A^2\|_F \leq \|A\|_2^{k-2}\|A^2\|_F \leq \|A\|_2^{k-2}\|A\|_F^2$. Therefore,

$$\left|\sum_{k\geq 3}\frac{(-1)^k}{k}\mathrm{tr}(A^k)\right| \ \leq \ \sum_{k\geq 3}\frac{1}{k}\,|\mathrm{tr}(A^k)| \ \leq \ \sqrt{n}\,\|A\|_F^2\sum_{k\geq 3}\frac{1}{k}\,\|A\|_2^{k-2} \ \leq \ \frac{\sqrt{n}}{3(1-\rho)}\,\|A\|_F^2\,\|A\|_2.$$

In particular, since $\|A\|_2 \leq \|A\|_F$ and $\|A\|_2 \leq \rho$, the remainder is bounded by $c\|A\|_F^3$ for a constant $c$ depending only on $n$ and $\rho$. Choose $\varepsilon > 0$ so that whenever $\|A\|_F \leq \varepsilon$, we have $c\|A\|_F \leq \frac{1}{4}$. Then for $\|C - I\|_F = \|A\|_F \leq \varepsilon$ we obtain $-\log\det C \geq \frac{1}{4}\|A\|_F^2$ and hence $\mathcal{R}_{\mathrm{KL}}(U) = -\frac{1}{2}\log\det C \geq \frac{1}{8}\|A\|_F^2 = \frac{1}{8}\|C - I\|_F^2$. □

### 3.7 Information-theoretic two-sided endpoint mechanisms

We give a *model-based* information-theoretic argument isolating two checkable *endpoint mechanisms* suggesting that extreme redundancy levels can be suboptimal in stylized settings within the framework of Definition 3.11. We present two mechanisms corresponding to common coordinate-wise product channels $K$: (i) when $K$ implements per-coordinate quantization at fixed resolution $\Delta > 0$ (so $\widetilde{Z}$ is discrete/quantized) under a marginal-entropy budget, large redundancy can force the *joint* entropy $H(\widetilde{Z})$ (and thus the information budget available to the task) to shrink; and (ii) when $K$ is (or includes) component-wise independent corruption, exact independence ($R = 0$) can be strictly improvable by adding an arbitrarily weak redundant copy (Appendix B). These are separate stylized scenarios; they do not, by themselves, assert that a single fixed $K$ simultaneously verifies both endpoint conditions of Lemma 3.17.

This capacity-side mechanism is stated in the fixed-resolution quantized (Shannon-entropy) setting of Convention 1.1, and is independent of the Gaussian approximation results in Section 3.2.

**Lemma 3.23** (Capacity-side information constraint). *Let $\widetilde{Z} = (\widetilde{Z}_1, \ldots, \widetilde{Z}_n)$ be the downstream representation. Assume $\widetilde{Z}$ is discrete or quantized at fixed resolution $\Delta > 0$ (Convention 1.1), e.g., $K_i = Q_\Delta$ for each coordinate, and $\sum_{i=1}^n H(\widetilde{Z}_i) \leq B_0$. If the redundancy coordinate is the total correlation $R = \mathcal{R}_{\mathrm{KL}}(\widetilde{Z}) = \mathrm{TC}(\widetilde{Z})$, then*

$$I(\widetilde{Z}; S) \ \leq \ H(\widetilde{Z}) \ = \ \sum_{i=1}^n H(\widetilde{Z}_i) \ - \ \mathrm{TC}(\widetilde{Z}) \ \leq \ B_0 - R. \tag{6}$$

*In particular, $R \leq B_0$ for all representations satisfying the entropy budget above. Under this entropy-budgeted model class one typically takes $R_{\max} \leq B_0$.*

*Proof.* Under Convention 1.1, the entropies are Shannon entropies (discrete or quantized) and are finite under the budget $\sum_i H(\widetilde{Z}_i) \leq B_0 < \infty$. For discrete random variables, the identity $H(\widetilde{Z}) = \sum_i H(\widetilde{Z}_i) - \mathrm{TC}(\widetilde{Z})$ holds by the definition of total correlation. (This identity does NOT hold for differential entropy, which is why we use Convention 1.1 to ensure all entropies are Shannon entropies.) The entropy budget gives $H(\widetilde{Z}) \leq B_0 - R$, and $I(\widetilde{Z}; S) \leq H(\widetilde{Z})$ yields equation 6. □

**Corollary 3.24** (High-redundancy endpoint is zero-information). *Under Convention 1.1 and the assumptions of Lemma 3.23, if a representation with $R = B_0$ is feasible (i.e., exists), then it satisfies $H(\widetilde{Z}) = 0$ and is therefore almost surely constant; hence $I(\widetilde{Z}; S) = 0$. In particular, if $R_{\max} = B_0$ and such a representation exists, then $R = R_{\max}$ implies $I(\widetilde{Z}; S) = 0$.*

**Zero-information baseline.** Define

$$\mathcal{E}_0 \ := \ \inf_{\widehat{S}:\ I(\widehat{S}; S)=0} \mathbb{E}\big[\ell_{\mathrm{task}}(S, \widehat{S})\big], \tag{7}$$

where the infimum is taken over all random variables $\widehat{S}$ (on the same probability space as $S$) such that $I(\widehat{S}; S) = 0$. This is the best achievable risk among predictors carrying no information about $S$ (e.g., the optimal constant predictor under squared loss). If $R_{\max} = B_0$ and the constraint $R = R_{\max}$ is feasible, then by Corollary 3.24,

$$\mathcal{E}(R_{\max}) \ \geq \ \mathcal{E}_0. \tag{8}$$

**Nontriviality (empirically checkable).** If

$$\inf_{R \in [0, R_{\max})} \mathcal{E}(R) \; < \; \mathcal{E}_0, \tag{9}$$

then the endpoint $R = R_{\max}$ is strictly suboptimal as a global minimizer. Condition equation 9 is directly checkable: it requires that some representation performs strictly better than the best zero-information predictor.

**Low-redundancy endpoint.** Proposition 3.14 provides a complementary mechanism: under component-wise independent corruption, $R = 0$ is strictly improvable by injecting an arbitrarily weak redundant copy, yielding lower task risk while $R = \mathcal{R}_{\mathrm{KL}}(\widetilde{Z}) \downarrow 0$ (Appendix B).

### 3.8 Interior redundancy regime induced by competing bounds

*Remark* 3.25 (Competing endpoint constraints motivate an interior regime). Let $\mathcal{E} : [0, R_{\max}] \to \mathbb{R}$ be lower semicontinuous and attained. In stylized corruption/capacity models, one can often establish opposing endpoint mechanisms: near $R = 0$, introducing a small amount of cross-coordinate coupling can improve robustness to component-wise corruption; near large $R$, large total correlation under a marginal-entropy budget forces $H(\widetilde{Z})$ and thus $I(\widetilde{Z}; S)$ to collapse. We record this as an interpretive heuristic—not as a global shape assumption on $\mathcal{E}(R)$—which motivates examining near-optimal representations in an interior range away from both extremes.

**Interpretation.** The takeaway is practical: it is informative to evaluate performance across a range of $R$ rather than only at an endpoint.

## 4 Experiments

**Goal.** We conduct controlled sweeps that relate downstream linear-probe performance to a *realized* redundancy coordinate and proxy diagnostics, with seed-based uncertainty.

**Setup.** We use masked autoencoders (MAE) He et al. (2022); after pretraining, the encoder is frozen and evaluated with a linear probe. Let $Z$ denote the frozen encoder representation *used as input to the linear probe*: specifically, we use global average pooling over patch tokens (excluding the CLS token) from the final encoder layer, yielding a 768-dimensional feature vector per image.

**Architecture and data.** We use a ViT-Base/16 backbone (patch size $16 \times 16$, embedding dimension 768, 12 layers, 12 heads) with a standard MAE decoder (embedding dimension 512, 8 layers). All experiments use CIFAR-10 for pretraining and in-domain evaluation, with the standard train/test split (50,000 training images, 10,000 test images); CIFAR-100 is used for transfer evaluation with its standard split. Pretraining runs for 50 epochs; linear probes are trained for 50 epochs.

**Pretraining hyperparameters.** We use the AdamW optimizer with learning rate $10^{-4}$, weight decay 0.05, and a cosine learning rate schedule with 3 warmup epochs (minimum learning rate $10^{-5}$). The per-GPU batch size is 128; gradient clipping is applied with max norm 1.0; mixed-precision training (AMP) is enabled. Data augmentation consists of resizing to $224 \times 224$, random horizontal flipping, and per-channel normalization to $[-1, 1]$ (mean and std both 0.5). These hyperparameters are held fixed across all E1 runs. We pretrain with a redundancy-regularized objective

$$\mathcal{L}_{\mathrm{total}} \; = \; \mathcal{L}_{\mathrm{recon}} \; + \; \lambda_{\mathrm{red}} \, \mathcal{R}_{\mathrm{spec}}(\cdot). \tag{10}$$

**Reconstruction loss implementation.** In our MAE implementation, $\mathcal{L}_{\mathrm{recon}}$ is the mean-squared reconstruction error averaged over *masked patches only* (75% masking), with per-patch pixel averaging in the patchified space; input pixels are scaled to $[-1, 1]$. This yields end-of-training MAE validation losses on the order of $10^{-2}$ in our runs.

**Training-time vs. analysis-time redundancy measurement.** During pretraining, the redundancy term is estimated on mini-batch *patch-token* features by flattening tokens within the batch (each patch token treated as a sample in the covariance estimate). For analysis and all reported redundancy coordinates, we compute the *realized* $\mathcal{R}_{\text{spec}}$ *post-hoc* on the frozen probe features $Z$ (one sample per image), matching the probe input. Unless stated otherwise, all figures and all tests that condition on redundancy use this realized coordinate on $Z$.

**Diagnostic coordinate vs. theory coordinate.** We regularize a tractable diagnostic ($\mathcal{R}_{\text{spec}}$) and do not claim it equals TC. The theoretical statements in Section 3.4 concern the KL/TC redundancy coordinate $R = \text{TC}(\widetilde{Z})$ on the downstream representation $\widetilde{Z} = K(\phi_\theta(X))$. Accordingly, we analyze outcomes as a function of a *realized redundancy coordinate* (measured diagnostics such as $\mathcal{R}_{\text{spec}}$ and local proxies such as $\|C - I\|_F^2$ and $\mathcal{R}_{\chi^2}$, and an estimate of TC when feasible), rather than by $\lambda_{\text{red}}$ directly (no monotonicity in $\lambda_{\text{red}} \mapsto R$ is assumed). We report reconstruction/validation loss, linear-probe performance, and redundancy diagnostics, including proxy-consistency checks.

**Downstream risk vs. reconstruction loss.** The theoretical profile $\mathcal{E}(R)$ concerns attainable *downstream task risk* as a function of a redundancy coordinate (Section 3.4), not MAE reconstruction or validation loss. Accordingly, our empirical conclusions are stated in terms of frozen-encoder linear-probe accuracy; reconstruction/validation loss is reported only as a training-stability diagnostic and to avoid misleading "all runs have nearly identical validation loss" interpretations.

**Validation-loss variability (not "nearly identical").** Across the $N = 33$ E1 runs (where $N$ denotes the number of runs), the end-of-training MAE validation loss varies meaningfully with $\lambda_{\text{red}}$ and seed: the overall range is min $= 0.02385$ to max $= 0.03490$ (about $46\%$ relative spread computed as max/min $- 1$; equivalently, the range is $\approx 38\%$ of the midpoint), and even within a fixed $\lambda_{\text{red}}$ group the seed-to-seed spread ranges from about $4\%$ to $11\%$ (computed as max/min $- 1$ across the three seeds). Thus, probe-accuracy differences should not be attributed to trivially "identical" reconstruction behavior.

**Proxy-consistency (falsifiable diagnostic check).** Because the theory is stated for $R = \text{TC}(\widetilde{Z})$ on $\widetilde{Z} = K(\phi_\theta(X))$ (Section 3.4), while practice relies on tractable diagnostics/proxies ($\mathcal{R}_{\text{spec}}$, $\|C - I\|_F^2$, $\mathcal{R}_{\chi^2}$), we treat strong proxy agreement as a necessary guardrail: if proxy-consistency is weak, the intended theory-to-measurement link fails for that setting and redundancy-sweep conclusions should be interpreted only in diagnostic terms. Proxy consistency is necessary (not sufficient) for the intended theory-to-measurement link.

**A computable bridge to total correlation in a computable regime.** To partially bridge the diagnostic coordinate to the KL/TC coordinate, we report a sanity check in a regime where TC is operationally computable from second-order statistics. Let $C$ denote the correlation matrix of the standardized frozen probe features $Z$; for a mean-zero Gaussian with correlation $C$,

$$\text{TC}_{\text{Gauss}}(Z) = -\tfrac{1}{2} \log \det(C).$$

We compute $\text{TC}_{\text{Gauss}}$ *post-hoc* on frozen $Z$ for all E1 runs using a stable signed log-determinant (`slogdet`) with a small diagonal ridge ($\epsilon = 10^{-6}$) for numerical stability. On the E1 runs, this Gaussian-TC proxy serves as a sanity check toward the KL/TC coordinate and is treated only as a diagnostic bridge. We treat this only as a sanity bridge (not an equivalence claim), since the Gaussian identity need not hold outside covariance-dominated regimes.

As an additional (optional) falsifiable check outside Gaussian assumptions, one can estimate a coarse TC on a small "sanity set" by: fixing a low-dimensional projection (e.g., $m \in \{4, 6, 8\}$ coordinates of the pooled probe features or a fixed random projection), discretizing each coordinate with a fixed binning rule (e.g., $B = 16$ equal-mass bins), and computing a plug-in estimate $\widehat{\text{TC}} = \sum_i \widehat{H}(U_i) - \widehat{H}(U)$ on a fixed sample budget. We treat such checks as guardrails against complete proxy–TC misalignment rather than as replacements for the theory coordinate.

### 4.1 Experimental Design

We run a controlled sweep of redundancy regularization strengths $\lambda_{\mathrm{red}}$ using MAE pretraining with the objective in equation 10, and evaluate the resulting representations with frozen-encoder linear probes. Our design uses 11 values of $\lambda_{\mathrm{red}}$ and 3 random seeds ($N = 33$ runs total), holding the backbone and training budget fixed. In this sweep we use 50 pretraining epochs and seeds $\{42, 43, 44\}$, holding all remaining hyperparameters fixed. Concretely, the grid is $\lambda_{\mathrm{red}} \in \{-0.01, 0, 0.001, 0.005, 0.01, 0.02, 0.05, 0.1, 0.2, 0.3, 0.5\}$. We do *not* advocate negative $\lambda_{\mathrm{red}}$ as a meaningful "regularization" setting; we include a single small negative value only as a diagnostic stress-test that encourages higher realized redundancy under the same pipeline.

For each run, we record downstream linear-probe accuracy on CIFAR-10 (in-domain) and CIFAR-100 (transfer) together with redundancy diagnostics computed on the frozen probe features $Z$. The goal is not to infer a universal functional form, but to test whether performance can be higher in an interior redundancy regime than at both endpoints, consistent with the endpoint-strictness logic used in Lemma 3.17. Throughout E1, the $\lambda_{\mathrm{red}}$ grid and all evaluation procedures are fixed *a priori*; in particular, the probe training protocol is held constant across runs (no per-$\lambda$ tuning), so differences are attributable to the learned representation rather than probe-side hyperparameter choices. Concretely, the linear head is trained with a fixed protocol (SGD with momentum 0.9, learning rate 0.1, cosine schedule, 50 epochs, no weight decay), and probe features are z-score standardized using training-split statistics.

**Why analyze performance versus a realized redundancy coordinate.** The sweep parameter $\lambda_{\mathrm{red}}$ is an optimization control knob, but it is not assumed to map monotonically to a single redundancy level across runs. Accordingly, we analyze outcomes as a function of the *realized redundancy coordinate* $\mathcal{R}_{\mathrm{spec}}$ computed on frozen $Z$ at the end of training, operationalized here as the logged statistic `red_spec_proxy`. Figure 2b illustrates that the same $\lambda_{\mathrm{red}}$ can yield different realized values across seeds, and that different $\lambda_{\mathrm{red}}$ settings can overlap in the realized coordinate. This motivates using realized redundancy as the organizing axis rather than treating $\lambda_{\mathrm{red}}$ as the final explanatory variable. To strengthen interpretability of this coordinate, we also compare `red_spec_proxy` against a covariance-based proxy `red_cov_proxy`; Figure 3b reports strong rank agreement (Spearman $\rho = 0.874$, $N = 33$), supporting the use of `red_spec_proxy` to order runs in subsequent analyses.

**Endpoint–interior inequality test (curve-shape free).** To align with the endpoint strictness logic in Lemma 3.17 while remaining robust to non-monotone or non-smooth behavior, we use an *endpoint–interior inequality*. Let $\Delta_{\mathrm{EI}}$ denote

$$\Delta_{\mathrm{EI}} := \Big(\text{best interior window mean accuracy}\Big) - \Big(\text{best endpoint mean accuracy}\Big),$$

where the *endpoint mean* is defined as the better of the low-$R$ endpoint regime and the high-$R$ endpoint regime, and each regime mean is computed over an equal-sized subset of runs at the corresponding extreme of the realized redundancy axis. The interior term is computed over an equal-sized contiguous window within the remaining (non-endpoint) runs, and we take the best interior window mean.

Concretely, we sort runs by realized redundancy and define the low/high endpoint regimes as the bottom/top ($k$) runs, with ($k$) fixed *a priori* (in our sweep, $k = 7$, roughly 20% of runs); we then scan all contiguous interior windows of width ($k$) within the remaining runs and take the best interior-window mean. (The choice $N = 33$, $k = 7$ admits at least one interior window under this rule.) The bootstrap procedure recomputes this same scanning rule, so uncertainty reflects the window-selection step. This test does *not* assume a smooth unimodal curve and does *not require* fitting a specific parametric model; it is an inequality-style check for whether an interior regime can outperform both endpoints. We estimate uncertainty via a nonparametric bootstrap at the run level: we resample runs with replacement (20,000 bootstrap replicates), recompute $\Delta_{\mathrm{EI}}$, and report a 95% bootstrap interval and $P(\Delta_{\mathrm{EI}} > 0)$.

### 4.2 Experimental Results

We summarize results using realized redundancy coordinates and an endpoint–interior inequality as primary evidence, without imposing a curve-fitting model. Figure 2b emphasizes that $\lambda_{\mathrm{red}}$ is best viewed as a control

parameter (with seed-dependent realized outcomes), while Figure 2a reorganizes the same runs by realized redundancy and shows the resulting performance regimes more directly.

**Training stability and diagnostic transients (why we rely on realized coordinates).** Figure 1 summarizes representative E1 training trajectories. Panels (A–B) show that MAE training and validation losses decrease smoothly with epoch, but also exhibit meaningful separation across $\lambda_{\text{red}}$, reinforcing that reconstruction/validation loss is a *training-stability* diagnostic rather than the theory-facing risk object. Panels (C–D) visualize *training-time* redundancy diagnostics logged on mini-batch patch-token features (tokens flattened within batch). While the spectral diagnostic in (C) behaves smoothly, the covariance proxy in (D) displays a systematic "rise-then-fall" pattern early in training across all shown $\lambda_{\text{red}}$ values. This behavior is explained by scale sensitivity: the proxy in (D) is computed from a *raw covariance* matrix (not correlation), hence its magnitude depends on feature variance. Indeed, the overlaid $\text{tr}(\Sigma)$ (dotted; right axis) increases sharply in the same early-epoch window, indicating a transient variance/scale transition as the encoder representations move from near-random to structured features; after this transition, redundancy control reduces off-diagonal structure and the proxy declines. Because such transients are intrinsic to training-time mini-batch estimation and can reflect feature-scale dynamics rather than the end-state redundancy ordering, all reported redundancy coordinates and all endpoint–interior analyses in E1 use the *analysis-time realized* diagnostics computed *post-hoc* on frozen probe features $Z$ (one sample per image), matching the probe input and avoiding scale artifacts.

**Qualitative regimes (organized by realized redundancy).** Across both probe tasks, low realized redundancy is associated with lower accuracy relative to intermediate values (Figure 2a). At higher realized redundancy, behavior is task-dependent: CIFAR-100 transfer accuracy is more sensitive and can be lower than the intermediate regime, whereas CIFAR-10 in-domain accuracy tends to saturate over a broad range. These observations motivate an inequality-based test of whether an interior regime can outperform both endpoints without assuming any particular global shape.

**Primary statistical evidence via endpoint–interior inequality.** Figure 3a reports the bootstrap distribution of $\Delta_{\text{EI}}$ for each task. For CIFAR-100 (transfer), the estimated improvement of the best interior regime over the better endpoint is $\Delta_{\text{EI}} = +0.0357$ with 95% CI $[+0.0252, +0.0409]$ and $P(\Delta_{\text{EI}} > 0) > 0.999$ (all 20,000 bootstrap replicates yielded $\Delta_{\text{EI}} > 0$; this is empirical frequency, not a formal probability bound). For CIFAR-10 (in-domain), the corresponding effect is smaller but still positive: $\Delta_{\text{EI}} = +0.0101$ with 95% CI $[+0.0031, +0.0147]$ and $P(\Delta_{\text{EI}} > 0) = 0.993$. The larger effect size for CIFAR-100 is consistent with sensitivity to task complexity and distribution shift in how redundancy impacts linear readout performance, though the magnitude may vary across settings.

**Operational estimate of an interior optimizer location (secondary).** To address requests for an explicit estimate of an "equilibrium" point, we additionally fit a quadratic model in the realized coordinate `red_spec_proxy` (pooled over all runs) and report the implied maximizer $\widehat{R}_{\text{fit}} = -b/(2a)$ when the fitted curvature satisfies $a < 0$. This estimate is reported as a descriptive summary in the realized diagnostic coordinate (not as a proof of shape, and not as a claim about the KL/TC coordinate without further proxy assumptions). For CIFAR-10 (in-domain), the fitted maximizer is $\widehat{R}_{\text{fit}} \approx 0.60$ with bootstrap 95% CI $[0.56, 0.64]$; for CIFAR-100 (transfer), $\widehat{R}_{\text{fit}} \approx 0.51$ with bootstrap 95% CI $[0.47, 0.54]$.

**Robustness to excluding the negative-$\lambda_{\text{red}}$ diagnostic point.** Because negative $\lambda_{\text{red}}$ inverts the intended penalty and is not advocated as a meaningful regularization setting, we also recompute the endpoint–interior inequality after excluding the three $\lambda_{\text{red}} = -0.01$ runs. With fixed $k = 7$, the transfer task remains strongly positive ($\Delta_{\text{EI}} \approx 0.033$ with $P(\Delta_{\text{EI}} > 0) \approx 0.999$), while the in-domain effect becomes weaker (CI can include 0 under the same $k$), suggesting that the clearest interior-regime evidence in E1 is driven by the transfer probe. Importantly, the quadratic-fit location estimates are essentially unchanged by excluding negative $\lambda_{\text{red}}$ (CIFAR-10 $\widehat{R}_{\text{fit}} \approx 0.60$, CIFAR-100 $\widehat{R}_{\text{fit}} \approx 0.51$), indicating that the point-estimate locations are not an artifact of the negative-$\lambda_{\text{red}}$ diagnostic point.

**Proxy consistency and interpretability of the realized coordinate.** Figure 3b shows that the realized redundancy ordering is broadly consistent across the spectral (`red_spec_proxy`) and covariance (`red_cov_proxy`) diagnostics (Spearman $\rho = 0.874$, $N = 33$), strengthening the validity of using `red_spec_proxy` as the organizing coordinate for Figures 2a–3a.

**E2 (VICReg): an attempted non-reconstruction SSL check and a validity-gate failure.** To probe whether the interior-regime advantage in E1 is MAE-specific, we conducted an additional sweep using VICReg (a non-reconstruction SSL objective) while keeping the *same* analysis protocol: (i) define the realized redundancy coordinate on the frozen encoder features used for probing, (ii) reparameterize outcomes by the realized coordinate rather than by the intervention parameter, and (iii) apply the endpoint–interior inequality test only when the realized coordinate exhibits sufficient regime coverage.

**VICReg sweep hyperparameters.** The VICReg sweep uses the same ViT-Base/16 encoder backbone as E1. We train using AdamW (learning rate $10^{-4}$, weight decay 0.05, cosine schedule with 3 warmup epochs), per-GPU batch size 256, and 50 epochs. The VICReg projector is a 3-layer MLP with hidden dimension 2048 and output dimension 2048, following the standard VICReg architecture. VICReg loss coefficients are $\alpha = 25$ (invariance), $\beta = 25$ (variance), and $\gamma = 1$ (covariance), as in the original VICReg paper. Data augmentation follows the standard VICReg protocol: RandomResizedCrop(224, scale=(0.2, 1.0)), RandomHorizontalFlip, ColorJitter (brightness/contrast/saturation/hue $= 0.4/0.4/0.2/0.1$, probability 0.8), RandomGrayscale (probability 0.2), and GaussianBlur (kernel size 9, sigma $\in [0.1, 2.0]$). The sweep grid varies $\lambda_{\mathrm{red}} \in \{0, 0.001, 0.005, 0.01, 0.02, 0.05, 0.1, 0.2\}$ (two seeds per setting; $N = 16$ runs total).

In the VICReg sweep, the intervention parameter $\lambda_{\mathrm{red}}$ is strongly rank-associated with the measured spectral diagnostic $\mathcal{R}_{\mathrm{spec}}$ (Spearman $\rho \approx -0.93$), and the spectral and covariance-based diagnostics remain mutually consistent (Spearman $\rho \approx 0.94$ between $\mathcal{R}_{\mathrm{spec}}$ and a covariance proxy). However, the realized redundancy coordinate shows *insufficient regime coverage*: across the sweep grid, $\mathcal{R}_{\mathrm{spec}}$ varies only over an approximately 1.7% relative range (max/min$-1$). We treat regime coverage as a *validity gate* for regime-based claims: if the realized coordinate does not traverse meaningfully distinct low/intermediate/high regimes, then endpoint–interior comparisons are underpowered and can be dominated by noise in both the coordinate and probe accuracy. Accordingly, the VICReg sweep does *not* satisfy our validity gate for applying the endpoint–interior inequality test, and we do not interpret the resulting probe-accuracy trends as evidence for or against an interior optimum in this coordinate.

This outcome is nonetheless informative for the broader experimental thesis: the slow/structural coordinate that mediates downstream generalization can be *objective-dependent.* That is, redundancy (as operationalized by $\mathcal{R}_{\mathrm{spec}}$ and related proxies) behaves as a controllable and regime-spanning coordinate in the MAE setting (E1), whereas under VICReg the same coordinate appears effectively constrained by the base SSL objective over the tested intervention range, preventing a regime-spanning test. This motivates a general experimental workflow: for a given foundation objective and chosen coordinate, establish (a) intervention–coordinate controllability, (b) regime coverage of the realized coordinate, and (c) proxy-consistency across diagnostics, before drawing endpoint–interior conclusions.

**Scope/Limitations.** These results are shown for MAE under a fixed backbone and training budget; extending to other self-supervised paradigms, network sizes (undercomplete vs. overcomplete regimes), or longer training is future work. Accordingly, we do not claim invariance of the preferred realized redundancy regime across model scales. We also do not interpret negative $\lambda_{\mathrm{red}}$ as a meaningful regularization knob: beyond a single diagnostic point, we do not pursue negative-$\lambda_{\mathrm{red}}$ sweeps because they invert the intended penalty and can introduce optimization/instability confounds. All E1 runs also fix generic regularizers/hyperparameters (optimizer, augmentation, and weight decay) so within-sweep comparisons are not confounded by changing those controls. A natural additional control would be to repeat a tiny subset of settings under a small weight-decay sweep while keeping all else fixed: e.g., run $\lambda_{\mathrm{red}} \in \{0, 0.05\}$ with weight decay $\in \{0.0, 0.05, 0.1\}$ over the same 3 seeds (18 runs total). This tests whether the redundancy–probe relationship can be reproduced by generic regularization alone.

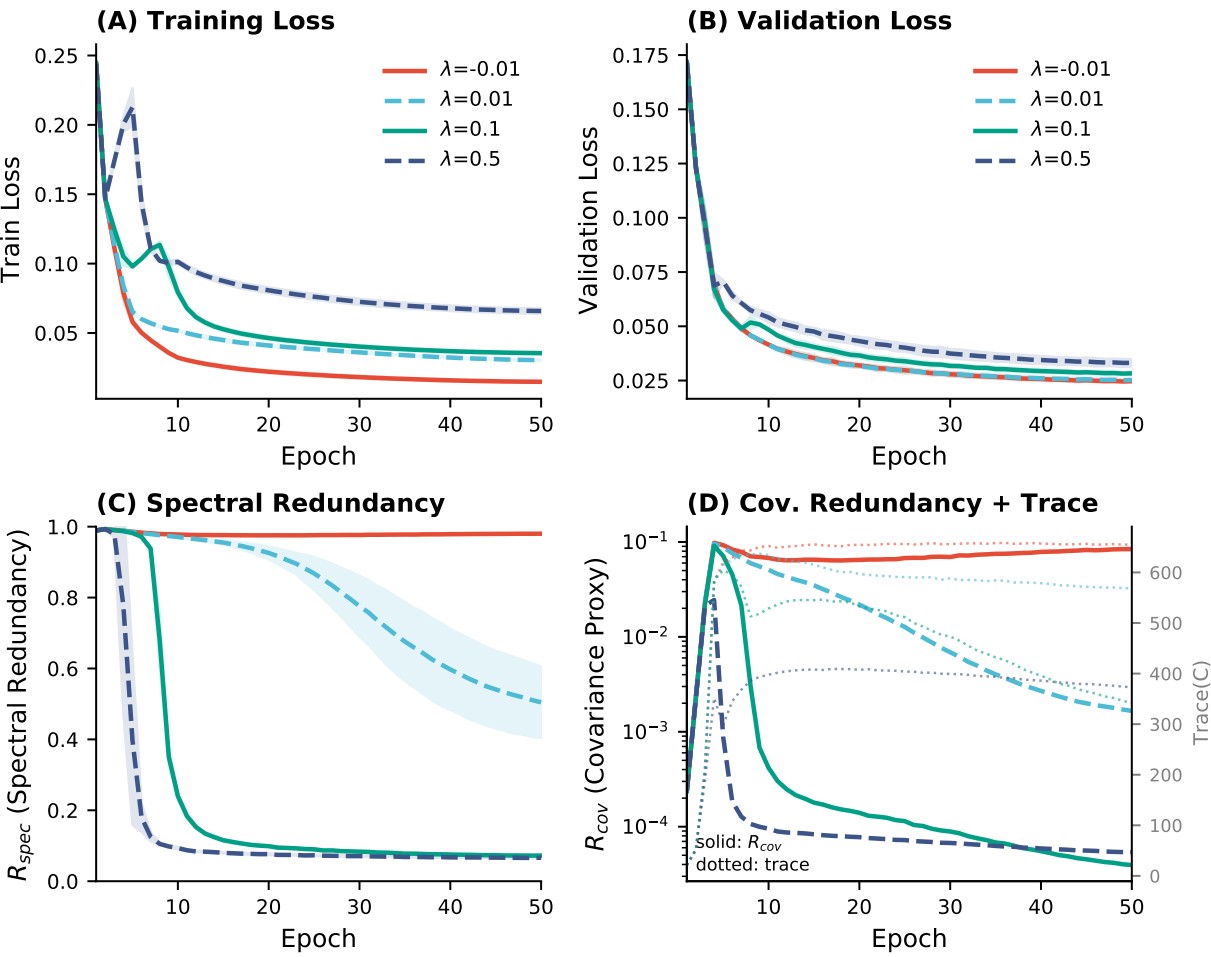

Figure 1: **E1 training dynamics and diagnostic transients (illustrative $\lambda_{\mathrm{red}}$ values).** Curves summarize MAE pretraining dynamics across seeds (mean $\pm$ std over seeds $\{42, 43, 44\}$) for representative regularization strengths $\lambda_{\mathrm{red}} \in \{-0.01, 0.01, 0.1, 0.5\}$. **(A) Training loss** and **(B) validation loss** decrease over epochs, with visible but nontrivial separation across $\lambda_{\mathrm{red}}$, consistent with the fact that reconstruction loss is *not* identical across runs and should not be treated as a proxy for downstream risk. **(C) Spectral redundancy diagnostic** (`red_spec_proxy`) and **(D) covariance-based proxy** (`red_cov_proxy`) are shown as *training-time* logged diagnostics computed on mini-batch patch-token features (flattened within batch). Importantly, panel (D) uses a *raw covariance* (not correlation) off-diagonal energy proxy and is therefore *scale-sensitive*. To diagnose this effect, we overlay the feature-scale statistic $\mathrm{tr}(\Sigma)$ (dotted curves; right axis), where $\Sigma$ is the same mini-batch covariance matrix used by the proxy computation. The characteristic "rise-then-fall" in the covariance proxy early in training aligns with a transient increase in $\mathrm{tr}(\Sigma)$, reflecting a variance/scale transition as representations move from near-random to structured features; the subsequent decline reflects reduced off-diagonal structure under redundancy control. Unless explicitly stated otherwise, our *reported* redundancy coordinates and all endpoint–interior tests use *analysis-time realized* diagnostics computed *post-hoc* on frozen probe features $Z$ (one sample per image), matching the probe input and avoiding training-time scale transients.

## 5  Discussion

**Summary.**  We define redundancy as dependence among representation coordinates via an $f$-divergence from independence (Definition 3.1), and in the KL special case use the total-correlation coordinate $R = \mathrm{TC}(\widetilde{Z})$ on the downstream representation $\widetilde{Z} = K(\phi_\theta(X))$ (Section 3.4). Under a well-posedness/attainment

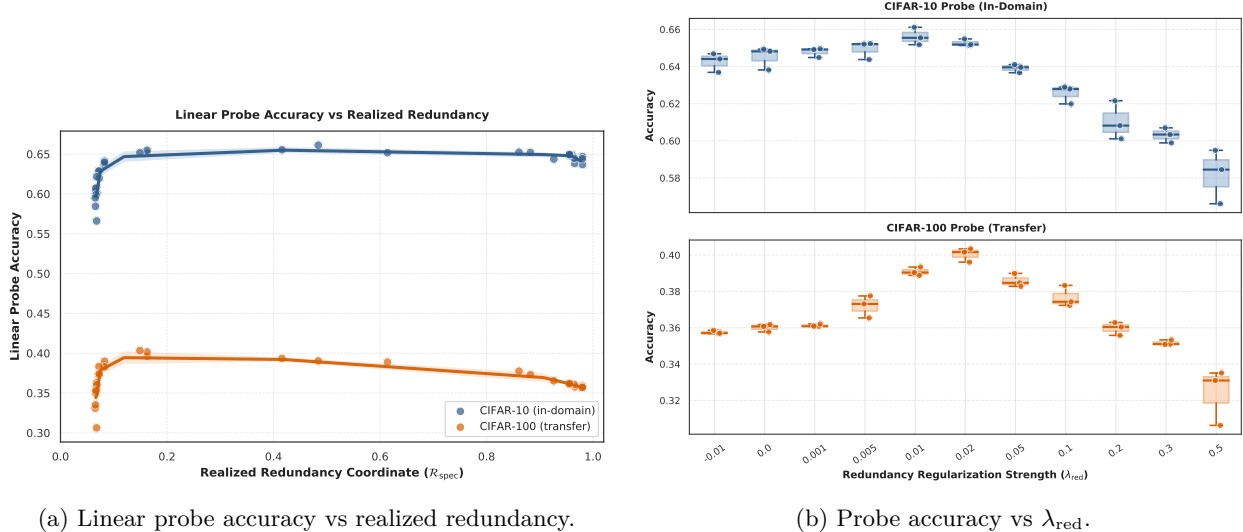

(a) Linear probe accuracy vs realized redundancy.

(b) Probe accuracy vs $\lambda_{\mathrm{red}}$.

Figure 2: Experimental overview for the $N = 33$ sweep runs. **(a)** Scatter over seeds with a nonparametric trend line for CIFAR-10 (in-domain) and CIFAR-100 (transfer) as a function of the realized redundancy coordinate `red_spec_proxy`. **(b)** The same runs grouped by the control parameter $\lambda_{\mathrm{red}}$ (boxplots with per-seed points), illustrating that $\lambda_{\mathrm{red}}$ is a control knob and that realized redundancy can overlap across settings.

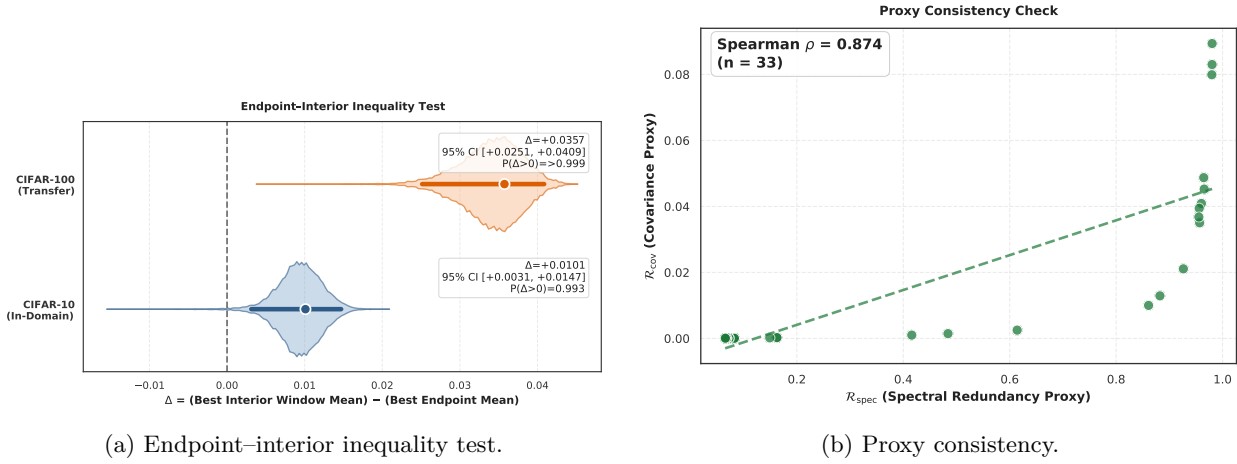

(a) Endpoint–interior inequality test.

(b) Proxy consistency.

Figure 3: Statistical evidence and proxy validation. **(a)** Bootstrap distribution of $\Delta_{\mathrm{EI}} = $ (best interior window mean accuracy)$-$(best endpoint mean accuracy), where the endpoint mean is the better of the low-$R$ and high-$R$ endpoint regimes. **(b)** Proxy consistency between `red_spec_proxy` and `red_cov_proxy`, supporting the realized redundancy ordering used in the analysis.

layer (Assumption 3.12) and endpoint-strictness inequalities (Lemma 3.17), the attainable downstream-risk profile $\mathcal{E}(R)$ admits at least one interior optimizer. Our main empirical finding is the corresponding *regime-level* result: in a controlled MAE sweep (E1) with fixed probe protocol, organizing runs by a realized redundancy diagnostic reveals an interior regime that outperforms both low-$R$ and high-$R$ endpoint regimes under an endpoint–interior inequality test (Figure 3a). We emphasize that this evidence does not rely on fitting a smooth unimodal curve, but on an inequality-style comparison aimed to reduce sensitivity to non-monotone behavior and seed noise. To support the intended measurement link, we report strong rank agreement between independent diagnostics (`red_spec_proxy` vs. `red_cov_proxy`; Figure 3b) and

include a sanity bridge toward the KL/TC coordinate in a computable regime via $\mathrm{TC}_{\mathrm{Gauss}} = -\frac{1}{2}\log\det(C)$ (Section 4.1).

**Scope and what is (and is not) claimed.** The theory concerns the KL/TC coordinate $R = \mathrm{TC}(\widetilde{Z})$, whereas the training-time intervention and analysis use tractable diagnostics such as $\mathcal{R}_{\mathrm{spec}}(Z)$ and local proxies. Accordingly, our empirical claim is *coordinate-conditional*: when the realized diagnostic is (i) controllable by the intervention, (ii) exhibits sufficient regime coverage, and (iii) is proxy-consistent with independent diagnostics, then an interior regime in that realized coordinate can be empirically identified as preferable for downstream linear probing. We do not claim that any particular diagnostic equals TC in general; proxy-consistency and sanity bridges are treated as guardrails rather than equivalences. Similarly, the endpoint mechanisms in Lemma 3.17 are offered as *witnesses* that endpoint strictness can arise from concrete structural reasons (e.g., corruption vs. capacity), not as a literal model of deep-net training.

**Limitations.** First, E1 is a controlled study in a single SSL paradigm (MAE) under a fixed backbone, budget, and pipeline; changing model scale (undercomplete vs. overcomplete), objective family, data regime, or training interventions can shift both the realized redundancy range and the preferred regime. Second, disentangling redundancy-specific effects from generic regularization remains important: although generic hyperparameters (including weight decay) are fixed within E1, a minimal additional control is to repeat a small subset of settings under a weight-decay sweep to test whether the redundancy–probe relationship can be reproduced by generic regularization alone. Third, the intervention variable $\lambda_{\mathrm{red}}$ is a control knob rather than a coordinate; since $\lambda_{\mathrm{red}} \mapsto R$ need not be monotone across seeds, we organize analyses by realized coordinates rather than asserting a direct causal mapping from $\lambda_{\mathrm{red}}$ to redundancy. Finally, negative $\lambda_{\mathrm{red}}$ is used only as a diagnostic stress-test and is not advocated as a meaningful regularization knob.

**Non-universality across objectives: an informative failure mode.** An attempted non-reconstruction SSL check with VICReg (E2) illustrates a concrete failure mode for regime-based tests: even when $\lambda_{\mathrm{red}}$ is rank-associated with the measured diagnostic and diagnostics remain mutually consistent, the realized coordinate can exhibit insufficient regime coverage, preventing a meaningful endpoint–interior comparison. This suggests that the practically relevant slow/structural coordinate mediating downstream generalization can be *objective-dependent*: a coordinate that is controllable and regime-spanning under one foundation objective (here, MAE) may be effectively constrained or saturated under another objective family over the same intervention range. This motivates a general experimental workflow: for a chosen coordinate and objective family, establish (a) intervention–coordinate controllability, (b) regime coverage in the realized coordinate, and (c) proxy-consistency across diagnostics before drawing endpoint–interior conclusions.

**Outlook: redundancy as a slow variable and beyond.** A promising direction is to treat redundancy (or, more generally, a structural diagnostic) as a *slow variable*: along training, many fast degrees of freedom (optimization noise, feature rotations, probe dynamics) may relax quickly relative to a slowly drifting structural coordinate. If such separation holds empirically, it suggests moving from static sweeps to *feedback control*: schedules or controllers that target a desired coordinate band by monitoring a realized diagnostic online, rather than selecting a fixed $\lambda_{\mathrm{red}}$. At the scale of foundation models, the key question becomes not whether a single curve is "U-shaped," but which coordinate is slow and controllable for a given objective family, how preferred regimes shift with scale/data/task distribution, and when diagnostics can be upgraded into reliable estimators of the KL/TC coordinate for theory-facing tests. Developing such estimators (or certifiable bridges in restricted regimes) and connecting coordinate control to predictable downstream generalization are natural next steps.

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

## A  Regularity of the attainable risk profile

We provide sufficient conditions under which the attainable risk profile $\mathcal{E}(R)$ in Definition 3.11 is well-defined, lower semicontinuous, and attains a minimum on a compact interval. These conditions are purely sufficient; for KL/TC coordinates in expressive model classes, additional regularity may be needed for the continuity assumptions to hold. They serve as an abstract well-posedness layer justifying Assumption 3.12; we do not claim that practical training dynamics satisfy these regularity conditions.

**Lemma A.1** (Sufficient conditions for attainment and lower semicontinuity of $\mathcal{E}$). *Let $\Theta$ and $\mathcal{G}$ be compact parameter spaces. Let $\mathcal{R} : \Theta \to \mathbb{R}$ be continuous, where $\mathcal{R}(\theta) \coloneqq \mathcal{R}(K(\phi_\theta(X)))$ is the redundancy coordinate induced by $\phi_\theta$ under the fixed channel $K$ in Definition 3.11.*

*Assume the task risk*

$$J(\theta, g) \coloneqq \mathbb{E}[\ell_{\text{task}}(S, g(K(\phi_\theta(X))))]$$

*is finite and continuous on $\Theta \times \mathcal{G}$ (e.g., by dominated convergence under a common integrable envelope for $\ell_{\text{task}}$). In particular, under these assumptions $\mathcal{R}(\theta) \in \mathbb{R}$ and $J(\theta, g) < \infty$ for all $(\theta, g) \in \Theta \times \mathcal{G}$, so every such pair is admissible in the sense of Definition 3.11.*

*Fix $R_{\max} > 0$ and $\varepsilon_0 > 0$. For $R \in [0, R_{\max}]$, define the feasible set*

$$\Gamma_{\varepsilon_0}(R) \coloneqq \{(\theta, g) \in \Theta \times \mathcal{G} : |\mathcal{R}(\theta) - R| \leq \varepsilon_0\}.$$

*Assume $\Gamma_{\varepsilon_0}(R)$ is nonempty for all $R \in [0, R_{\max}]$. Define the value function*

$$\mathcal{E}(R) := \inf_{(\theta,g)\in\Gamma_{\varepsilon_0}(R)} J(\theta, g).$$

*Then $\mathcal{E}$ is lower semicontinuous on $[0, R_{\max}]$ and attains a global minimum on $[0, R_{\max}]$.*

*Proof.* Since $\Theta \times \mathcal{G}$ is compact and $(\theta, g) \mapsto |\mathcal{R}(\theta) - R|$ is continuous, the correspondence $R \mapsto \Gamma_{\varepsilon_0}(R)$ has nonempty compact values and a closed graph (hence is upper hemicontinuous). Since $J$ is continuous, Berge's maximum theorem (applied to the minimization problem) implies the value function $R \mapsto \inf_{(\theta,g)\in\Gamma_{\varepsilon_0}(R)} J(\theta, g)$ is lower semicontinuous on $[0, R_{\max}]$ and the infimum is attained for each $R$. Since $[0, R_{\max}]$ is compact, $\mathcal{E}$ attains a global minimum. □

## B  A low-redundancy constructive example

This appendix provides a complete derivation for Proposition 3.14. The key point is local: in a simple linear-Gaussian model with component-wise independent corruption, exact independence in the KL redundancy coordinate ($\mathcal{R}_{\mathrm{KL}}(\widetilde{Z}) = D_{\mathrm{KL}}(P_{\widetilde{Z}}\|\Pi_{\widetilde{Z}}) = 0$) is strictly improvable by adding an arbitrarily weak redundant copy, which reduces the attainable squared error under optimal linear decoding while $\mathcal{R}_{\mathrm{KL}}(\widetilde{Z}) \downarrow 0$.

### B.1  Proof of Proposition 3.14

Assume $n \geq d + 1$ and define, for $\alpha \in (0, 1]$,

$$Z_0 = (S_1, \ldots, S_d, 0, \ldots, 0) \in \mathbb{R}^n, \qquad Z_\alpha = (S_1, \ldots, S_d, \alpha S_1, 0, \ldots, 0) \in \mathbb{R}^n.$$

Both are realizable under the gain constraint by taking rows of $W$ equal to $e_1^\top, \ldots, e_d^\top, \alpha e_1^\top, 0, \ldots, 0$.

Since $\mathrm{Cov}(S)$ is diagonal and $S$ is Gaussian, the coordinates of $Z_0$ are independent. Therefore $\widetilde{Z}_0 = Z_0 + \xi$ has independent coordinates as well, hence $\mathcal{R}_{\mathrm{KL}}(\widetilde{Z}_0) = 0$.

For $\widetilde{Z}_\alpha = Z_\alpha + \xi$, the only dependence is between

$$Y_1 = S_1 + \xi_1, \qquad Y_2 = \alpha S_1 + \xi_{d+1},$$

and all remaining coordinates are independent of $(Y_1, Y_2)$ and mutually independent; hence $\mathrm{TC}(\widetilde{Z}_\alpha) = \mathrm{TC}(Y_1, Y_2)$, where $(Y_1, Y_2)$ are jointly Gaussian with correlation

$$\rho(\alpha) = \frac{\alpha \nu_1}{\sqrt{(\nu_1 + \sigma^2)(\alpha^2 \nu_1 + \sigma^2)}}.$$

The total correlation of this bivariate Gaussian equals

$$\mathcal{R}_{\mathrm{KL}}(\widetilde{Z}_\alpha) = \mathrm{TC}(\widetilde{Z}_\alpha) = \mathrm{TC}(Y_1, Y_2) = -\tfrac{1}{2}\log\!\left(1 - \rho(\alpha)^2\right) \xrightarrow[\alpha\downarrow 0]{} 0.$$

Under joint Gaussianity and squared loss, the optimal linear decoder coincides with the LMMSE estimator. The only affected coordinate is $S_1$, for which the decoder effectively observes two noisy measurements

$$Y_1 = S_1 + \xi_1, \qquad Y_2 = \alpha S_1 + \xi_{d+1},$$

with independent noise variance $\sigma^2$. The corresponding posterior variance is

$$\mathrm{Var}(S_1 \mid Y_1, Y_2) = \left(\nu_1^{-1} + \tfrac{1+\alpha^2}{\sigma^2}\right)^{-1},$$

which is strictly smaller than the baseline $\left(\nu_1^{-1} + \tfrac{1}{\sigma^2}\right)^{-1}$ for every $\alpha > 0$. All other coordinates of $S$ are unchanged, hence the overall optimal risk is strictly smaller for $Z_\alpha$ than for $Z_0$ while $\mathcal{R}_{\mathrm{KL}}(\widetilde{Z}_\alpha) \downarrow 0$. This proves Proposition 3.14.

# C  Additional proofs and technical lemmas

## C.1  Preliminaries and Conventions

**Notation link to the main text.** In this appendix, $U$ denotes a generic random vector for the redundancy functional $\mathcal{R}_f(U) = D_f(P_U \| \Pi_U)$; in the main text, $U$ typically corresponds to a learned representation such as $Z$ or its corrupted version $\widetilde{Z}$.

We use the $f$-divergence notation as in Definition 3.1. We use standard facts about $f$-divergences, including nonnegativity and data processing (Csiszár–Morimoto; DPI).

Throughout, $\Pi_U = \bigotimes_{i=1}^n P_{U_i}$ denotes the product of the marginals of $P_U$. We write $p$ for the joint density of $P_U$ and $p_i$ for that of $P_{U_i}$. For any symmetric matrix $A$, we denote by $\|A\|_F$ its Frobenius norm and by $\log \det(A)$ the (scalar) natural logarithm of $\det(A)$ for $A \in \mathrm{Sym}^{++}$ (equivalently, $\log \det(A) = \mathrm{tr}(\log A)$ where $\log A$ is the principal matrix logarithm).

## C.2  Proof of Proposition 3.3 (Nonnegativity and Identity of Indiscernibles)

*Proof.* By Definition 3.1, if $P_U \ll \Pi_U$ and $L = \frac{dP_U}{d\Pi_U}$, then

$$\mathcal{R}_f(U) = D_f(P_U \| \Pi_U) = \int_{\mathcal{U}} f(L(u)) \, d\Pi_U(u) = \mathbb{E}_{\Pi_U}[f(L)].$$

Since $f$ is convex and $f(1) = 0$, Jensen's inequality gives

$$\mathcal{R}_f(U) = \mathbb{E}_{\Pi_U}[f(L)] \geq f(\mathbb{E}_{\Pi_U}[L]) = f(1) = 0,$$

using $\mathbb{E}_{\Pi_U}[L] = \int \frac{dP_U}{d\Pi_U} \, d\Pi_U = 1$. If $P_U \not\ll \Pi_U$, then by Definition 3.1 we have $\mathcal{R}_f(U) = +\infty$, so nonnegativity also holds.

If, in addition, $f$ is strictly convex on some interval containing 1 and $\mathcal{R}_f(U) = 0$ with $P_U \ll \Pi_U$, then equality holds in Jensen's inequality, which forces $L = 1$ $\Pi_U$-a.e. Hence $P_U = \Pi_U$. Conversely, if $P_U = \Pi_U$, then $L \equiv 1$ and $\mathcal{R}_f(U) = 0$. $\qquad\square$

## C.3  Proof of Proposition 3.4 (Data Processing Inequality)

*Proof.* Let $K = \bigotimes_{i=1}^n K_i$ denote the product Markov kernel mapping probability measures on $U$ to those on $Y$, where each $K_i$ acts on coordinate $U_i$. Then $P_Y = P_U K$ and, since $K$ factorizes coordinate-wise, $\Pi_Y = \Pi_U K$. By the data processing inequality for $f$-divergences,

$$\mathcal{R}_f(Y) \;=\; D_f(P_Y \| \Pi_Y) \;=\; D_f(P_U K \| \Pi_U K) \;\leq\; D_f(P_U \| \Pi_U) \;=\; \mathcal{R}_f(U),$$

which establishes the claim. $\qquad\square$

## C.4  Proof of Proposition 3.5 (Bounds)

*Proof.* **(1) Lower bound.** This follows directly from Proposition 3.3.

**(2) Upper bound under bounded likelihood ratio.** Assume $P_U \ll \Pi_U$ and let $L = \frac{dP_U}{d\Pi_U}$. If $m \leq L \leq M$ $\Pi_U$-a.e., then $f(L) \leq \sup_{t \in [m,M]} f(t)$ $\Pi_U$-a.e., hence

$$\mathcal{R}_f(U) = \mathbb{E}_{\Pi_U}[f(L)] \;\leq\; \sup_{t \in [m,M]} f(t) \;<\; \infty.$$

**(3) KL kernel.** For $f(t) = t \log t$, the map $t \mapsto t \log t$ is continuous (indeed convex) on $[m, M]$, so it attains its maximum on this compact interval at an endpoint. Therefore

$$\mathcal{R}_{\mathrm{KL}}(U) = \mathbb{E}_{\Pi_U}[L \log L] \;\leq\; \max\{m \log m, \; M \log M\}.$$

Moreover, since $L \leq M$ implies $L \log L \leq L \log M$ (with the convention $0 \log 0 := 0$) and $\mathbb{E}_{\Pi_U}[L] = 1$, we also have the bound $\mathcal{R}_{\mathrm{KL}}(U) = \mathbb{E}_{\Pi_U}[L \log L] \leq \log M$. **Sanity check.** Since $\mathbb{E}_{\Pi_U}[L] = 1$, the bound $L \leq M$ $\Pi_U$-a.e. implies necessarily $M \geq 1$ (otherwise $\mathbb{E}_{\Pi_U}[L] \leq M < 1$), and similarly $m \leq 1$ when $m \leq L$ $\Pi_U$-a.e. $\qquad\square$

## C.5 Proof of Proposition 3.6 (Gaussian Total Correlation)

*Proof.* Let $U \sim \mathcal{N}(0, \Sigma)$ with correlation matrix

$$C = \mathrm{diag}(\Sigma)^{-1/2} \, \Sigma \, \mathrm{diag}(\Sigma)^{-1/2}.$$

Since $\Sigma \succ 0$, all diagonal entries $\Sigma_{ii} > 0$, so the diagonal matrix $\mathrm{diag}(\Sigma)$ (with diagonal entries from $\Sigma$) is invertible and $\mathrm{diag}(\Sigma)^{-1/2}$ is well-defined. Then $P_U = \mathcal{N}(0, \Sigma)$ and $\Pi_U = \bigotimes_i \mathcal{N}(0, \Sigma_{ii})$. The KL divergence between these two Gaussian measures is given by the standard Gaussian KL formula. Note that $\Pi_U = \bigotimes_i \mathcal{N}(0, \Sigma_{ii})$ is the multivariate Gaussian $\mathcal{N}(0, D)$ where $D = \mathrm{diag}(\Sigma_{11}, \ldots, \Sigma_{nn})$ is the diagonal matrix of marginal variances. Applying the standard formula $D_{\mathrm{KL}}(\mathcal{N}(0, \Sigma_1) \| \mathcal{N}(0, \Sigma_2)) = \frac{1}{2}(\mathrm{tr}(\Sigma_2^{-1} \Sigma_1) - n - \log \det \Sigma_1 + \log \det \Sigma_2)$ with $\Sigma_1 = \Sigma$ and $\Sigma_2 = D$, and using $\mathrm{tr}(D^{-1}\Sigma) = \sum_i \Sigma_{ii}/\Sigma_{ii} = n$, we obtain

$$D_{\mathrm{KL}}(\mathcal{N}(0, \Sigma) \,\|\, \mathcal{N}(0, D)) \;=\; \frac{1}{2}\Big(n - n - \log \det \Sigma + \sum_i \log \Sigma_{ii}\Big) \;=\; \frac{1}{2}\Big(\sum_i \log \Sigma_{ii} - \log \det \Sigma\Big).$$

Using the factorization $\Sigma = \mathrm{diag}(\Sigma)^{1/2} C \mathrm{diag}(\Sigma)^{1/2}$, we obtain

$$\log \det \Sigma = \sum_i \log \Sigma_{ii} + \log \det C,$$

and therefore

$$D_{\mathrm{KL}}(P_U \| \Pi_U) \;=\; -\tfrac{1}{2} \log \det C,$$

which proves Proposition 3.6. $\qquad\square$

## C.6 Proof of Proposition 3.7 (Quadratic Approximation and Covariance Form)

*Proof.* This subsection provides exact formulas and controlled expansions for the jointly Gaussian case.

**Setup and standardization.** Let $U \sim \mathcal{N}(0, \Sigma)$ and let $C = \mathrm{corr}(U)$ be its correlation matrix. After coordinate-wise standardization to unit marginal variances (i.e., replacing $U$ by $\mathrm{diag}(\Sigma)^{-1/2}U$), the joint law becomes $P = \mathcal{N}(0, C)$ and the product-of-marginals law becomes $Q = \mathcal{N}(0, I)$. Since both KL divergence and Pearson $\chi^2$ divergence are invariant under invertible linear changes of variables (a standard invariance property of $f$-divergences under bijections), it suffices to work with $(P, Q) = (\mathcal{N}(0, C), \mathcal{N}(0, I))$.

**(ii) Quadratic ($\chi^2$) redundancy: closed form.** Let $p$ and $q$ denote the densities of $P$ and $Q$ w.r.t. Lebesgue measure on $\mathbb{R}^n$. The likelihood ratio is

$$\frac{p(x)}{q(x)} = |C|^{-1/2} \exp\Big(-\tfrac{1}{2} x^\top (C^{-1} - I)x\Big).$$

By definition,

$$\chi^2(P\|Q) = \int \Big(\frac{p}{q} - 1\Big)^2 dQ = \int \frac{p(x)^2}{q(x)} \, dx - 1,$$

where we used $\int \frac{p}{q} \, dQ = \int p \, dx = 1$. Compute the Gaussian integral:

$$\int \frac{p(x)^2}{q(x)} \, dx = |C|^{-1} (2\pi)^{-n/2} \int \exp\Big(-x^\top(C^{-1} - I)x - \tfrac{1}{2}x^\top x\Big) dx$$

$$= |C|^{-1} (2\pi)^{-n/2} \int \exp\Big(-\tfrac{1}{2}x^\top(2C^{-1} - I)x\Big) dx.$$

This integral is finite iff $2C^{-1} - I \succ 0$, equivalently $2I - C \succ 0$. In particular, a sufficient condition is $\|C - I\|_2 < 1$ (equivalently $\|A\|_2 < 1$ with $C = I + A$), which implies $2I - C = I - A \succ 0$. In that case,

$$(2\pi)^{-n/2} \int \exp\left( - \tfrac{1}{2} x^\top (2C^{-1} - I) x \right) dx = |2C^{-1} - I|^{-1/2},$$

and hence

$$\int \frac{p(x)^2}{q(x)} \, dx = |C|^{-1} \, |2C^{-1} - I|^{-1/2} = |C|^{-1/2} \, |2I - C|^{-1/2}.$$

Therefore, under $2I - C \succ 0$ (so $\chi^2(P\|Q)$ is finite),

$$\chi^2(P\|Q) = |C|^{-1/2} \, |2I - C|^{-1/2} - 1. \tag{11}$$

For the Pearson kernel $f_{\chi^2}(t) = \tfrac{1}{2}(t-1)^2$, we have $D_{f_{\chi^2}}(P\|Q) = \tfrac{1}{2}\chi^2(P\|Q)$, so

$$\mathcal{R}_{\chi^2}(U) = D_{f_{\chi^2}}(P_U\|\Pi_U) = \tfrac{1}{2}\chi^2(P\|Q) = \tfrac{1}{2}\Big( |C|^{-1/2} \, |2I - C|^{-1/2} - 1 \Big).$$

**Controlled second-order expansion.** Write $C = I + A$ with $\|A\|_2 < 1$, so $I \pm A \succ 0$ and in particular $2I - C = I - A \succ 0$. Using equation 11 and $|2I - C| = |I - A|$,

$$|C|^{-1/2} |2I - C|^{-1/2} = |I + A|^{-1/2} |I - A|^{-1/2} = \exp\left( \tfrac{1}{2}\big[ - \log\det(I+A) - \log\det(I-A) \big] \right).$$

Because $C$ is a correlation matrix, $\mathrm{diag}(C) = \mathbf{1}$ and hence $\mathrm{diag}(A) = 0$; in particular $\mathrm{tr}(A) = 0$ and $\mathrm{tr}(A^2) = \|A\|_F^2$.

For $\|A\|_2 < 1$, the matrix-log series converges and

$$- \log\det(I \pm A) = -\mathrm{tr}\log(I \pm A) = \sum_{k \geq 1} \frac{(\mp 1)^k}{k} \mathrm{tr}(A^k) = \mp \mathrm{tr}(A) \; + \; \tfrac{1}{2}\mathrm{tr}(A^2) \; + \; R_\pm,$$

where the remainder satisfies the bound

$$|R_\pm| \leq \sum_{k \geq 3} \frac{1}{k} |\mathrm{tr}(A^k)| \leq \sum_{k \geq 3} \frac{1}{k} \|A\|_2^{k-2} \mathrm{tr}(A^2) \leq \frac{\|A\|_F^2 \, \|A\|_2}{3(1 - \|A\|_2)} = O(\|A\|_F^3),$$

using $\|A\|_2 \leq \|A\|_F$ and $\mathrm{tr}(A^2) = \|A\|_F^2$. With $\mathrm{tr}(A) = 0$, we therefore have

$$- \log\det(I \pm A) = \tfrac{1}{2}\|A\|_F^2 + O(\|A\|_F^3).$$

Substituting into the exponent yields

$$\tfrac{1}{2}\big[ - \log\det(I+A) - \log\det(I-A) \big] = \tfrac{1}{2}\|A\|_F^2 + O(\|A\|_F^3),$$

and exponentiating gives (for sufficiently small $\|A\|_F$, e.g. $\|A\|_F \leq 1$)

$$|I+A|^{-1/2}|I-A|^{-1/2} = \exp\left( \tfrac{1}{2}\|A\|_F^2 + O(\|A\|_F^3) \right) = 1 + \tfrac{1}{2}\|A\|_F^2 + O(\|A\|_F^3).$$

Substituting into $\mathcal{R}_{\chi^2}(U) = \tfrac{1}{2}(|I+A|^{-1/2}|I-A|^{-1/2} - 1)$ yields

$$\mathcal{R}_{\chi^2}(U) = \tfrac{1}{4}\|A\|_F^2 + O(\|A\|_F^3) = \tfrac{1}{4}\|C - I\|_F^2 + O(\|C - I\|_F^3).$$

**(i) KL / total-correlation redundancy: exact identity and expansion.** In the standardized Gaussian setting $(P, Q) = (\mathcal{N}(0, C), \mathcal{N}(0, I))$, the KL divergence has the exact closed form

$$D_{\mathrm{KL}}(P\|Q) = \tfrac{1}{2}\big( \mathrm{tr}(C) - \log\det C - n \big) = -\tfrac{1}{2}\log\det C,$$

since $C$ is a correlation matrix and $\text{tr}(C) = n$. This is Proposition 3.6. With $C = I + A$ and $\|A\|_2 < 1$, we have

$$-\log\det(I+A) = -\text{tr}\log(I+A) = \sum_{k \geq 1} \frac{(-1)^k}{k}\,\text{tr}(A^k).$$

Because $\text{tr}(A) = \text{tr}(C - I) = 0$ and $\text{tr}(A^2) = \|A\|_F^2$, the remainder satisfies

$$\left|\sum_{k \geq 3}\frac{(-1)^k}{k}\,\text{tr}(A^k)\right| \leq \sum_{k \geq 3}\frac{1}{k}\,|\text{tr}(A^k)| \leq \sum_{k \geq 3}\frac{1}{k}\,\|A\|_2^{k-2}\,\text{tr}(A^2) \leq \frac{\|A\|_F^2\,\|A\|_2}{3(1 - \|A\|_2)} = O(\|A\|_F^3),$$

using $\|A\|_2 \leq \|A\|_F$. Therefore,

$$\mathcal{R}_{\text{KL}}(U) = -\tfrac{1}{2}\log\det C = \tfrac{1}{4}\,\|A\|_F^2 + O(\|A\|_F^3) = \tfrac{1}{4}\,\|C - I\|_F^2 + O(\|C - I\|_F^3),$$

as claimed. $\qquad\square$

## C.7  Proof of Definition/Proposition 3.8–3.9 (Spectral Redundancy)

*Proof.* Let $\lambda_1, \ldots, \lambda_n$ be the eigenvalues of the correlation matrix $C$, so $\sum_{j=1}^n \lambda_j = \text{tr}(C) = n$, and define the normalized spectrum

$$\tilde{\lambda}_i \;=\; \frac{\lambda_i}{\sum_{j=1}^n \lambda_j}, \qquad \sum_{i=1}^n \tilde{\lambda}_i = 1.$$

The spectral entropy is

$$H_\lambda(Z) \;=\; -\sum_{i=1}^n \tilde{\lambda}_i \log \tilde{\lambda}_i,$$

which satisfies $H_\lambda(Z) \in [0, \log n]$, with $H_\lambda(Z) = \log n$ if and only if $\tilde{\lambda}$ is uniform and $H_\lambda(Z) = 0$ if and only if the spectrum is rank one. Setting $r_{\text{eff}}(Z) = \exp(H_\lambda(Z)) \in [1, n]$ and

$$\mathcal{R}_{\text{spec}}(Z) \;=\; 1 - \frac{r_{\text{eff}}(Z)}{n},$$

we obtain

$$\mathcal{R}_{\text{spec}}(Z) \in \left[0,\, 1 - \tfrac{1}{n}\right].$$

The lower and upper extremes correspond, respectively, to the uniform (maximally spread) and rank-one (completely collapsed) spectra. These bounds follow directly from standard entropy inequalities. $\qquad\square$

## C.8  Proof of Lemma 3.21 (Spectral Norm Control)

*Proof.* Let $Z = WX$ with $\|W\|_2 \leq M$ and $\mathbb{E}\|X\|^2 \leq \sigma^2$.

$$\text{tr}(\Sigma_Z) = \mathbb{E}\|Z - \mathbb{E}Z\|^2 = \mathbb{E}\|W(X - \mathbb{E}X)\|^2 \leq \|W\|_2^2\,\mathbb{E}\|X - \mathbb{E}X\|^2 \leq \|W\|_2^2\,\mathbb{E}\|X\|^2 \leq M^2\sigma^2.$$

Therefore, the total spectral energy of $Z$ is bounded, implying that the normalized eigenvalue spectrum is well defined and that $r_{\text{eff}}(Z) \in [1, n]$. Consequently, $\mathcal{R}_{\text{spec}}(Z) \in [0,\, 1 - \tfrac{1}{n}]$ by Definition 3.8, completing the proof. $\qquad\square$

## C.9  Proof of Lemma 3.22 (KL vs. Frobenius Near Independence)

The proof is provided inline following the lemma statement in Section 3.6.

### C.10 Proof of Lemma 3.23 (Capacity-Side Bound)

*Proof.* By the entropy decomposition,

$$H(\widetilde{Z}) = \sum_i H(\widetilde{Z}_i) - \mathrm{TC}(\widetilde{Z}) = \sum_i H(\widetilde{Z}_i) - \mathcal{R}_{\mathrm{KL}}(\widetilde{Z}).$$

Under the assumed entropy budget $\sum_i H(\widetilde{Z}_i) \leq B_0$, this yields

$$H(\widetilde{Z}) \leq B_0 - \mathcal{R}_{\mathrm{KL}}(\widetilde{Z}).$$

Since mutual information is bounded by entropy, $I(\widetilde{Z}; S) \leq H(\widetilde{Z})$, increasing redundancy necessarily reduces the maximal mutual information between the representation $\widetilde{Z}$ and the task variable $S$. This yields equation 6 and completes the proof. □

