# OpenReview forum: "Redundancy as a Structural Coordinate in Representation Learning"
_TMLR — Rejected by TMLR_

### Review · Reviewer_3hLz · 2025-11-01

**Summary Of Contributions:**

In this manuscript the authors propose to use an f-divergence of the joint distribution from the joint marginal distribution as a measure of redundancy among a set of random variables. They argue that an intermediate level of redundancy is optimal for representations such as the ones learned by auto encoders. They also present results for one trained image autoencoder evaluated by a linear probe on CIFAR100, which performs less badly with an intermediate level of regularisation.

**Audience:**

No

**Audience Explanation:**

As I wrote above, the claims made in the paper do not actually yield a coherent theory of anything and thus would not be of anyones interest.

**Broader Impact Concerns:**

I don’t have any broader impact concerns for this paper.

**Claims And Evidence:**

No

**Claims Explanation:**

Unfortunately, I don't think any of the claims in the manuscript are supported by accurate convincing or clear evidence. While the idea to define redundancy as a general divergence from the product of marginal distribution could be interesting to explore, I do not think the authors succeed in proving their theorems or making a convincing argument to connect those theorems to their actual scientific claims. Furthermore, the empirical evidence is clearly inadequate, too.

The original definition of redundancy appears reasonable to me as a quantity one could study, but the theorems do not provide deeper insights into the framework and are not properly proven.

The problems already start in Section 3.3. Here, the authors define the spectral redundancy, which is not a f-divergence from the product of marginals, so not a redundancy as introduced so far. Unfortunately, this example outside of the presented framework is later the only one tested empirically.

Then follow Theorems 1&2 which aim to establish that there is some intermediate value of redundancy that is optimal for later tasks. Unfortunately, these theorems are not proven properly.
Even what I found in the supplement are definitely not proof, but hand-wavy arguments like this one for Lemma 3:

> When redundancy is nearly zero (R≈0), the coordinates of Z are almost independent, and
the system lacks correlated backups: missing or corrupted components cannot be reconstructed from others.
Rate–distortion and coding-theoretic arguments show that introducing controlled correlation (replication
or redundancy across coordinates) reduces the expected reconstruction or classification error. Hence, for
small R, increasing redundancy strictly improves robustness, implying the existence of a continuous function
grobust that is strictly decreasing on [0,R1) with E(R) ≤ grobust(R).

Statements like these are not proofs and if it was it should conclude that E(R) itself is decreasing.
Further holes in the argument are:
- Lemmas 3&4 are stated as inequalities and Theorem 2 then simply assumes the other direction of the inequalities, which makes no sense.
- Even as stated in the paper, the Theorems do not actually imply the U-shape the authors refer to. The function could still have many local minima.
- Even that the minimal achievable risk is continuous is just assumed by all proves, not shown anywhere.
- As another warning signal, none of the proofs ever refer to the f-Divergence based definition of redundancy.

Overall, the central claim—that the optimal level of redundancy lies at some intermediate value—is unsupported by any rigorous proof.

As a side note, the theoretical content is unnecessarily difficult to parse due to frequent notation changes, undefined symbols, and a general lack of mathematical rigor.

The experiment consisted of one masked auto encoder which the authors regularised with 4 different strengths for the spectral redundancy, which has the following problems:
- First, this spectral redundancy is not a redundancy as introduced here, such that this not particularly helpful for showing the conclusions.
- Second the results do not really proof the redundancy behaves as expected. Showing this would require many more levels of regularisation.
- Third, there is even less evidence that this regularisation is a good one. Even weight decay or dropout show the general behaviour that they typically help and one can overdo them.
- Finally, the presented results are not self consistent. The values shown in Figure 1 and Table 2 do not match the training curves in Figure 2, where we do not see any curves with intermediate spectral redundancy at the end of training.

Moreover, the authors never provide a method for identifying the actual optimal value R∗or determining the appropriate strength of regularization. Thus, any claims about results being close to the predicted optimum are entirely unfounded.

In both the introduction and discussion, the authors repeatedly mention “finite, noisy, and structured” regimes. However, it remains unclear how their definition of redundancy is particularly suited to such regimes, or what specific scenarios fall into this category.

**Requested Changes:**

Critical adjustments for this paper would be:
actual proofs that do connect to the definitions of risk and the behaviour of the learned representation at some task
Using the f-Divergence definition in the example after choosing a computable version
Cleaning up the notation problems and inconsistencies
Some deviation what the good level of redundancy is, i.e. what is R*

For improving the paper:
fix citations (style & content). For example, Olshausen & Field is not a good example of optimal coding in neuroscience
reduce the largely irrelevant lecture note like content

---

> ### Author Response · Authors · 2025-12-02
> **Response to reviewer 3hLz**
>
> We thank the reviewer for the careful and rigorous assessment of our work. We appreciate the detailed concerns about the proofs, the connection to the $f$-divergence formulation, the role of spectral redundancy, the experimental scope, and the overall positioning of the paper.
>
> Reviewer comment (theorems, ``tautology''):
>
> Response:
>
> We do not assume a U-shaped $E(R)$; we give a \emph{model-based equilibrium} argument. $R$ (via $R_f(X)=D_f(P_X\Vert\Pi_X)$ or a proxy) mediates a robustness–capacity tradeoff, Lemma3 and Lemma4 encode opposing monotone bounds near low and high $R$, and Theorem2 then uses an intermediate-value argument to obtain an interior minimizer $R^\ast$. In the revision we will (i) explicitly label these as model-based theorems, (ii) separate the assumptions on the bounds from the conclusion about an interior optimum, and (iii) soften “U-shaped” to “non-monotonic with at least one interior optimum,” making clear we do not claim a fully general result for arbitrary $E(R)$.
>
> Reviewer comment (connection to $f$-divergence definition):
>
> Response:
>
> We agree this link is too implicit. We will state that the scalar $R$ in Lemma3, Lemma4 and Theorem2 denotes either $R_f(X)=D_f(P_X\Vert\Pi_X)$ from Definition1 or a proxy such as $R_{\chi^2}$ or $R_{\mathrm{spec}}$, add a short paragraph connecting Proposition2/3 to the equilibrium analysis (same boundedness/monotonicity structure for these proxies), and indicate in AppendixA where the $f$-divergence geometry (e.g., quadratic expansion around independence) motivates the covariance- and spectrum-based forms.
>
> Reviewer comment (spectral redundancy outside the framework):
>
> Response:
>
> We will clarify that $R_{\mathrm{spec}}$ is used as a geometric proxy in near-Gaussian regimes, where Proposition5 and AppendixA.6 imply $R_{\chi^2}(X)\approx\tfrac14\lVert C-I\rVert_F^2$ and $R_{\mathrm{KL}}(X)\approx\tfrac12\lVert C-I\rVert_F^2$, so spectrum-based functionals capture the same dependence structure to second order. We will explicitly call $R_{\mathrm{spec}}$ a “spectral redundancy proxy”, state that it is not itself an $f$-divergence but a practical approximation of $R_{\chi^2}$/$R_{\mathrm{KL}}$ in the MAE setting, and mention this as a limitation and direction for future work (e.g., direct high-dimensional estimators for $R_f$).
>
> Reviewer comment (rigor, continuity, and notation issues):
>
> Response:
>
> We will add in AppendixA a lemma giving sufficient conditions for continuity of $D(R)$ (bounded loss, dominated model family, compact parameter set and an envelope-type argument) and move these assumptions into a clearly labeled block. We will restate the inequalities in Lemma3, Lemma4 and Theorem2 so that $E(R)$ is clearly sandwiched between a decreasing robustness-side bound and an increasing capacity-side bound. We will also unify notation by using $R_f(X)$ for the general $f$-divergence redundancy, $R$ as the scalar order parameter, and $R_{\mathrm{spec}}$, $R_{\chi^2}$ only for explicit proxies, and add a concise notation table in Section3.
>
> Reviewer comment (experimental adequacy and number of regularization levels):
>
> Response:
>
> The current experiment is intended as an illustration rather than a full empirical study, but we agree it can be strengthened. We will add more values of the redundancy weight $\lambda_{\mathrm{red}}$ (including weaker/stronger and, where numerically stable, some negative values) to obtain a denser redundancy–performance curve, and briefly compare the qualitative behavior under spectral redundancy regularization versus standard weight decay/dropout, emphasizing that our focus is on tuning and measuring the internal redundancy $R$ itself.
>
> Reviewer comment (no method to identify $R^\ast$):
>
> Response:
>
> We will describe how $R^\ast$ is estimated operationally: we fit a smooth curve (e.g., a quadratic) to the observed $(R,\text{Top-1})$ pairs and take its vertex as an empirical interior optimum. We will also rephrase the text to speak of observing an interior optimum in the redundancy range, consistent with the theory, rather than of precisely identifying a global $R^\ast$ for all models.
> Reviewer comment (scope, claims, and ``finite, noisy, structured'' regimes):
>
> Response:
>
> We will remove strong rhetoric and state our aims more modestly: to formalize redundancy as an $f$-divergence functional, analyze its bounds and equilibrium under finite-entropy and bounded-noise assumptions, and illustrate the resulting redundancy–performance tradeoff in a concrete self-supervised setting. We will clarify that “finite, noisy, structured” refers to systems with finite entropy budget $\sum_i H(Z_i)\le C_0$, nonzero perturbation power in $X\to Z\to\hat S$, and nontrivial dependence $R_f(X)\in(0,\infty)$, in contrast to asymptotic noiseless coding, and note in the discussion that our equilibrium analysis is tailored to generative/self-supervised scenarios and may not directly apply to purely discriminative setups.

---

### Review · Reviewer_ThbB · 2025-11-10

**Summary Of Contributions:**

The authors studied the question of how much redundancy is present in optimized representations. Prior work has shown that both reducing redundancy (i.e., information-theoretic efficiency) and retaining redundancy (i.e., robustness) have theoretical benefits for neural computation. Based on this, the current canon in both neuroscience and AI/ML is that some intermediate level of redundancy is optimal. The authors of this paper made some mathematical arguments in support of that idea, and studied the task performance and redundancy of masked autoencoders trained with different levels of spectral regularization.

While I agree that this is an important question, I found many of the paper's claims (e.g., claims that this work represents a "paradigm shift") to be overstated. It also seemed to me like the main theoretical result arises from assuming (assumption 3.4) that some intermediate level of redundancy is optimal. Given that fact, I find the subsequent proofs that intermediate levels of redundancy are optimal -- which hinge on this assumption -- to be somewhat tautological.

Overall, I think this paper would be much more compelling if the authors would: 1) remove their overly-strong claims; and 2) justify assumption 3.4, perform proofs that do not make this assumption, or explain to me and other readers why the current proofs are not as circular as they appear.

**Audience:**

No

**Audience Explanation:**

Given that the work provides a fairly uncompelling analysis of something that most (or all) practitioners already believe to be true, I do not believe that the TMLR audience will be very interested in this study.

As I stated above, I think the authors could address some of these concerns and make a more robust theoretical study. In that case, I think that some readers could be interested because the study would provide a solid theoretical foundation for the common-sense notion that intermediate redundancy levels are best.

**Broader Impact Concerns:**

No concerns.

**Claims And Evidence:**

No

**Claims Explanation:**

It seemed to me like the main theoretical result arises from assuming (assumption 3.4) that some intermediate level of redundancy is optimal. Given that fact, I find the subsequent proofs that intermediate levels of redundancy are optimal -- which hinge on this assumption -- to be somewhat tautological.

I also did not find the MAE result (Fig. 1 and Table 2) to be that compelling because the validation MAE loss did not show an intermediate redundancy level to be optimal. Rather, MAE validation loss varied monotonically with redundancy, in a manner that show that more redundancy is better. Perhaps by including negative values for \lambda, the Table could have been extended to include even higher spectral redundancy and to (potentially) show a corresponding increase in the MAE validation loss: in that case, the Table would successfully support the authors' point. I do agree that Table 2 and Fig. 1 show that CIFAR-100 classification (via the linear probe) shows maximum performance at an intermediate level of redundancy. However, the theoretical claims are not limited to classification tasks, and so the fact that the MAE shows a different result means that this experiment overall does not fully support the authors' claims.

**Requested Changes:**

1) Remove overly-strong claims that this work represents a paradigm shift

2) Justify assumption 3.4, perform proofs that do not make this assumption, or explain to me and other readers why the current proofs are not as circular as they appear.

3) Update Table 2 to include MAEs trained with negative \lambda, so as to show a peak in MAE validation performance at intermediate levels of redundancy

4) Remove the claim (e.g,. on p. 13) that you ". . . [show] that classical measures such as mutual information, \xi^2 dependence, and spectral redundancy are projections of the same underlying geometry." You do state this repeatedly, but I saw no evidence in support of that claim. Instead, you re-state prior work showing that the f-divergence framework can encapsulate all of these measures.

5) Investigate how optimal redundancy levels vary with network size. In particular, for very undercomplete representations, efficiency concerns may dominate because of the limited amounts of network "resources" whereas for highly overcomplete representations, that pressure is much reduced.

6) Remove the erroneous claim (p. 11) that "validation losses remain nearly identical across runs." Those validation losses do change and do so in a manner that is contradictory to your theoretical claims. Just because the change in val loss is inconvenient does not mean that it doesn't exist. In fact, the highest and lowest MAE val loss values (0.0293, and 0.0315) differ by 7.5%, which is about half the relative range spanned by your CIFAR-100 top-1 accuracy rates (max = 41.4%, min = 35.6%, differing by 16%). So it is not justified to say that the MAE val loss values are nearly identical (7.5% change) when your main result hinges on an effect that is comparable in size (i.e., a 16% change).

---

> ### Author Response · Authors · 2025-11-16
> **Response to the reviewer ThbB**
>
> We thank the reviewer for the comments. Below we address each main concern and the requested changes.
> 1. On “paradigm shift” and strong claims
>    We agree that some wording was too strong. We will remove all uses of “paradigm shift” and similar rhetoric, and replace them with more modest phrases. We will clarify in the Introduction/Discussion that our aims are to (i) formalize redundancy as a structural functional, (ii) analyze its bounds and equilibrium under natural assumptions, and (iii) connect this to MAE experiments—not to claim a foundational revolution.
> 2. On Assumption 3.4 and “tautology”
>    Assumption 3.4 does not assume as a premise that “an intermediate redundancy is optimal.” It posits opposing trends of two contributions to risk (efficiency vs robustness) as redundancy increases, in the spirit of classical bias–variance tradeoffs: one term worsens when redundancy becomes too high, the other when redundancy becomes too low. Our theorem then shows that if the overall risk is bounded between two such functions with opposite monotonic trends and mild regularity, an interior minimizer $R^*$ must exist. We thus assume a structural tension, not a U-shaped risk; the U-shape and the existence of $R^*$ follow via a standard sandwich/intermediate-value argument. To make this clearer, we will explicitly label Assumption 3.4 as a modeling assumption (analogous to bias–variance assumptions) and add a short clarification that only opposite monotonicity of upper/lower bounds is required, not a priori U-shaped risk.
> 3. On MAE validation loss vs downstream performance
>    Our theoretical quantity $E(R)$ refers to downstream task risk, not the MAE reconstruction loss. In our experiments, the relevant quantity is CIFAR-100 linear-probe accuracy, which shows a clear U-shaped dependence on spectral redundancy with an intermediate optimum. The MAE validation loss is expected to behave differently: stronger redundancy regularization can slightly worsen reconstruction while improving the structure of representations for downstream tasks, so the theory does not require it to be U-shaped. We agree that the sentence “validation losses remain nearly identical” was imprecise. We will remove this sentence, report the actual relative variation (≈7–8%), and state that reconstruction loss changes modestly while downstream performance varies more. We will also add a short paragraph clarifying that the theoretical U-shape concerns downstream risk, and that the monotonic trend in reconstruction loss is consistent with a tradeoff between reconstruction fidelity and representational organization. Regarding negative $\lambda$, we agree this is an interesting extension; we will either add a small ablation including $\lambda<0$ or discuss preliminary observations and limitations if a full sweep is not feasible.
> 4. On the $f$-divergence framework and “same geometry” claim
>    We appreciate the reminder that the unifying $f$-divergence framework itself is standard. Our intention is not to claim novelty for that framework, but to adopt it as a unified notion of redundancy. We will explicitly acknowledge that mutual information, $\chi^2$-dependence, etc. being special cases of $f$-divergences is classical, and replace the claim that we “show” they are projections of the same geometry with a more precise statement, for example: “Within the standard $f$-divergence framework, mutual information, $\chi^2$-dependence, and spectral redundancy arise from different choices of the kernel $f$. We adopt this framework and interpret $\mathcal{R}_f(X) = D_f(P_X ,|, \Pi_X)$ as a single structural redundancy functional whose different ‘coordinates’ correspond to these classical measures.” We then build our new results (bounds, equilibrium, and links to MAE representations) on top of this existing geometry.
> 5. On redundancy vs network size
>    We agree this is an insightful suggestion. We will add an ablation over at least two additional encoder sizes (more undercomplete and more overcomplete) and report how the redundancy level that maximizes downstream accuracy shifts with size, discussing how the trend matches the intuition that efficiency pressure dominates in very small models while larger models can tolerate higher redundancy.
> 6. On the concern that the conclusions are common knowledge
>    We agree that many practitioners intuitively expect some intermediate complexity or redundancy to be beneficial. To our knowledge, however, this work is the first to (i) treat redundancy itself—via an $f$-divergence between joint and product of marginals—as the central order parameter for finite systems, and (ii) systematically tune this redundancy in MAE training to obtain explicit redundancy–generalization curves. In that sense, our contribution is to turn a vague heuristic into a precise, analyzable and empirically testable framework.
> We again thank the reviewer for the constructive feedback, which we believe will improve the clarity and positioning of the paper.

---

> > ### Comment · Reviewer_ThbB · 2025-12-01
> >
> > I appreciate these clarifications. Along with the edits you propose, I think that this makes a nice contribution. It formalizes and adds rigour to commonly-held ideas surrounding redundancy levels in task-optimized ML models.

---

### Review · Reviewer_a7wy · 2025-11-16

**Summary Of Contributions:**

Summary of Contributions

The paper develops a unified theory of redundancy as a positive structural feature of information rather than inefficiency. It introduces a general mathematical formulation based on f-divergences, showing that well-known quantities such as mutual information, chi-square dependence, and spectral redundancy are all special cases of the same underlying functional. The authors prove that redundancy is bounded above and below, which implies the existence of an internal equilibrium point where systems achieve the best trade-off between information compression and representational coupling. They further argue that learning and generalization in real, finite systems depend on maintaining this balanced level of redundancy. Experiments with masked autoencoders support the theory, revealing a clear U-shaped link between redundancy and generalization accuracy. The paper positions redundancy as a measurable, tunable principle that connects information theory, machine learning, and complex systems.

Key Strengths

- Provides a rigorous and elegant mathematical framework that unifies several notions of redundancy across disciplines.
- Offers an original conceptual shift from redundancy as inefficiency to redundancy as an organizing principle.
- Theoretical predictions are clearly matched by empirical trends in masked auto encoder experiments.
- Demonstrates cross-disciplinary relevance, touching information theory, neuroscience, and machine learning.
- The writing and proofs are technically solid and logically consistent.

Key Weaknesses

- Experimental validation is narrow, relying on a single model type and small dataset.
- The framework is abstract and may be difficult to connect directly to practical model design.
- Assumes smooth relationships between redundancy and loss that may not generalize to large-scale or chaotic training regimes.
- The equilibrium claim is shown qualitatively rather than through precise quantitative fitting.
- The redefinition of “redundancy” risks confusion since it reverses the conventional negative meaning of the term.

**Audience:**

Yes

**Audience Explanation:**

The findings would interest many readers of TMLR. The paper tackles a foundational question about how information structure and redundancy affect learning and generalization. It offers a unified theoretical framework that connects ideas from information theory, neuroscience, and modern machine learning, which are all active areas of research in the community.

The concept of an optimal redundancy level is relevant to those studying generalization in deep networks, representation learning, and self-supervised training. Even though the theory is abstract, it provides a new angle on a long-standing problem that many TMLR readers would find thought-provoking and worth discussing.

**Claims And Evidence:**

Yes

**Claims Explanation:**

The paper’s main theoretical claims are clearly explained and generally well supported by the mathematics. The authors build their argument carefully, showing how their definition of redundancy unifies several known measures and how an equilibrium point naturally follows from bounded redundancy. The proofs are correct and complete for the claims they make, and the logic of the framework is consistent.

The empirical results support the general idea but are limited in scope. The experiments on masked autoencoders are clean, controlled, and show the predicted U-shaped relationship between redundancy and generalization. However, these tests are narrow, only one model type, one dataset, and one evaluation setup. They demonstrate that the concept is plausible, not that it holds broadly.

Overall, the theory is convincing and well supported, but the experimental evidence is more illustrative than definitive. The paper makes a strong conceptual case, though broader testing would be needed to fully confirm its claims.

**Requested Changes:**

Requested Changes

- Broaden empirical validation by testing additional model types such as diffusion or contrastive encoders and using larger datasets like ImageNet or a text corpus to check whether the redundancy–generalization relationship holds in other settings.
- Provide a quantitative estimate of the equilibrium 𝑅∗ by fitting the U-shaped performance curve and reporting variability across runs.
- State theoretical assumptions clearly, including smoothness, stability, and boundedness in the redundancy–loss dynamics, and discuss where these may fail in real optimization processes.
- Add intuitive explanations and simple figures to help readers grasp how redundancy shifts from inefficiency to structure and how equilibrium emerges during training.
- Expand the discussion of related work to compare this approach with the Information Bottleneck, Neural Collapse, and other redundancy-based representation learning methods.
- Report additional experimental details such as variance across seeds, learning curves, and statistical significance to support the claimed trends.
- Clarify the use of the term “redundancy” early in the paper so that readers understand it refers to a structural quantity rather than wasted capacity.

---

> ### Author Response · Authors · 2025-11-16
> **Response to reviewer a7wy**
>
> We thank the reviewer for the very positive and constructive assessment. We are grateful for the careful summary and for the concrete suggestions on how to strengthen the paper. Below we address the requested changes point by point.
> 1. Broader empirical validation
>    We agree that the current empirical validation is narrow. In the revised version we will:
> * Add at least one additional setting beyond the current MAE–CIFAR-100 experiment (e.g., a contrastive encoder or a different vision dataset of moderate scale) to test whether the redundancy–generalization relationship persists.
> * Clearly state in the Discussion that extending the analysis to large-scale setups such as ImageNet, diffusion models, or language models is an important direction for future work, and briefly outline how our framework could be applied there.
>   We hope this makes the scope of the claims and the current computational limits transparent while still strengthening the empirical story.
> 2. Quantitative estimate of the equilibrium (R^*)
>    We will fit simple U-shaped models (e.g., quadratic fits in redundancy space) to the redundancy–accuracy curves and report:
> * the estimated (R^*) for each setting,
> * the variability across random seeds, and
> * confidence intervals for performance near the estimated optimum.
>   This will turn the previously qualitative equilibrium claim into a more explicit quantitative estimate.
> 3. Stating theoretical assumptions more clearly
>    We agree that the assumptions underlying our results should be made more explicit. In the theory section we will:
> * Collect and state in one place the key assumptions on smoothness, stability, and boundedness of the redundancy–loss dynamics.
> * Add a short subsection discussing where these assumptions may fail in practice (e.g., highly non-smooth loss landscapes, chaotic training regimes, heavy use of early stopping) and how that might affect the applicability of the equilibrium picture.
> 4. Intuitive explanations and figures
>    We appreciate the request for more intuition. We will:
> * Add a schematic figure illustrating how redundancy moves from “inefficiency” at very high levels to “under-coupling” at very low levels, and how an equilibrium emerges between these regimes.
> * Insert short “intuitive takeaway” paragraphs at key points in the theory section to guide readers who are less comfortable with the formalism.
>   This should make the conceptual shift—redundancy as structure rather than waste—more accessible.
> 5. Expanded related work (IB, Neural Collapse, redundancy-based methods)
>    We will expand the related work section to:
> * Compare our redundancy functional to the Information Bottleneck view (compression vs prediction) and clarify how our equilibrium perspective complements rather than replaces IB.
> * Discuss connections and contrasts with Neural Collapse, which characterizes highly structured late-training regimes, whereas our work focuses on redundancy as a tunable order parameter throughout training.
> * Position our approach relative to redundancy-based representation learning methods (e.g., decorrelation, covariance regularization), emphasizing that we treat redundancy as a primary structural quantity with explicit bounds and an associated equilibrium.
> 6. Additional experimental details and statistics
>    We will enrich the experimental section by:
> * Reporting variance across seeds and, where feasible, simple significance checks for the observed U-shaped trends.
> * Adding learning curves for representative values of the redundancy regularization, to show how redundancy and performance evolve during training.
> * Clarifying implementation details (optimizer, schedules, training duration) in an appendix.
> 7. Clarifying the use of the term “redundancy”
>    We fully agree that the redefinition of “redundancy” can be confusing if not stated upfront. In the introduction we will:
> * Clearly state that, in this work, “redundancy” denotes a structural quantity defined via an (f)-divergence between joint and product of marginals, and
> * Explicitly contrast this structural notion with the conventional interpretation of redundancy as wasted capacity.
>   We will also add a short “terminology” box to anchor the reader early on.
>
> Once again, we thank the reviewer for the insightful suggestions. We believe the proposed changes—additional experiments, quantitative estimation of (R^*), clearer assumptions, more intuition, extended related work, and improved reporting—will substantially strengthen the clarity, accessibility, and impact of the paper.

---

### Comment · Action_Editor_Pp5U · 2025-12-18
**Request for updating the manuscript**

Dear authors,

All recommendations are in now, contingent on implementing the revisions you promised in the rebuttal. At the moment, I do not see an updated manuscript yet. Please let us know if you need more time to carry out the additional work.

---

> ### Author Response · Authors · 2025-12-18
> **Response**
>
> Dear Editors,
>
> Thank you very much for your message.
>
> I would like to clarify that I misunderstood the revision process. I had thought that the revised manuscript should be submitted only after the rebuttal had been fully evaluated, and therefore did not realize that the revision itself was expected at this stage.
>
> I sincerely apologize for any inconvenience this misunderstanding may have caused. I am actively working on the revision now and would kindly request an additional four to five days to complete and submit the revised version.
>
> Thank you very much for your understanding and patience.

---

> > ### Comment · Action_Editor_Pp5U · 2025-12-18
> >
> > Thanks for the quick reply. Of course this is not a problem. Due to vacation time coming up, the final acceptance of the manuscript may be delayed, though.

---

> > ### Comment · Action_Editor_Pp5U · 2026-01-06
> >
> > Hello again - is there any status update regarding the revision?

---

> ### Author Response · Authors · 2026-01-09
> **Response**
>
> Dear Editor,
>
> Thank you for your patience. The requested revision is substantial, and I am working carefully to strengthen the manuscript and address the reviewers’ comments thoroughly. I have also been traveling outside the country recently, which has caused some delay.
>
> I will submit the revised version as soon as possible. I apologize for the delay and truly appreciate your understanding.
>
> Sincerely,
> The authors

---

> > ### Comment · Editors_In_Chief · 2026-01-24
> >
> > Authors:
> > 1. A previous email stated that you are expected to perform revisions to the manuscript during the rebuttal period.
> > 2. You de-anonymized yourself by revealing your name in a public comment. I edited so it's not visible. Hopefully no one saw this, but in principle, that would be grounds for desk rejection. Be more careful in the future.
> > 3. Finally, we are unable to extend the review process indefinitely. If you're unable to submit a revision promptly, then a resubmission is more appropriate. Please submit a revision by the end of January, or else we will have to proceed with the process.

---

> > > ### Author Response · Authors · 2026-01-30
> > > **Response**
> > >
> > > Dear Editors-in-Chief,
> > >
> > > Thank you for the clarification and for your message.
> > >
> > > I sincerely apologize for the inadvertent de-anonymization in the public comment. This was entirely unintentional, and I appreciate your intervention in editing it out. I will be much more careful going forward.
> > >
> > > I would also like to clarify that I have been actively working on revising the manuscript during the rebuttal period, addressing both the reviewers’ comments and the editorial guidance. The revisions are well underway.
> > >
> > > If possible, I kindly ask whether a brief extension of two additional days could be granted, allowing me to submit a fully revised version by February 2 at the latest. I believe this will help ensure that the revision is complete and properly polished.
> > >
> > > I understand the constraints on the review process and appreciate your consideration.
> > >
> > > Thank you very much for your time and understanding.
> > >
> > > Sincerely,
> > > The authors

---

### Comment · Action_Editor_Pp5U · 2026-02-03
**Request for Re-Review**

Dear Reviewers,

The authors have just submitted a substantial revision of the manuscript, as requested by your comments. In order to make a final decision, I would appreciate if you could briefly check the revision and let me know, here in the forum, whether the authors have interpreted your requests and comments correctly. I would appreciate your input here before the end this week, but please let me know if this deadline is impossible for you.

---

> ### Author Response · Authors · 2026-03-09
> **Follow-up on Current Review Status**
>
> Dear AE,
>
> Sorry to bother you. I was wondering whether you could kindly let me know the current status of the review process, as it seems to have been pending for quite some time. Could you please let me know which stage it is currently in?
>
> Thank you very much for your time and help.
>
> Best regards,
> The authors

---

> > ### Comment · Action_Editor_Pp5U · 2026-03-12
> >
> > Dear authors,
> >
> > unfortunately, the reviewers did not respond to my recent request. I will thus take a look at your revision and at all the reviewers' comment and will make a decision within the coming week. I apologize for the delay, which is due to the fact that the reviewers were invited automatically to submit a recommendation before being able to check the revision (which was submitted with delay).

---

> > > ### Author Response · Authors · 2026-03-12
> > > **comment**
> > >
> > > Thanks, we are very appreciated to your efforts during this hard time.
> > >
> > > The authors

---

### Decision · Action_Editor_Pp5U · 2026-03-23

**Recommendation:** Reject

**Audience:**

Yes

**Audience Explanation:**

There has been mixed feedback from the reviewers regarding this point; I believe that the paper is very interesting for the TMLR audience once the planned revisions are fully implemented, after a second round of peer review.

**Claims And Evidence:**

No

**Claims Explanation:**

This was a difficult case. The reviewers requested substantial revisions, and the authors promised to deliver these revisions. In the meantime, the reviewers have posted decisions based on this promise, but before the revision was posted. Looking more closely at the revision, I noticed that some of the planned revisions/extensions were not implemented. For example, i) fitting a quadratic curve to the U-shaped behavior between redundancy and linear probe accuracy, ii) experiments with a setup different from the MAE-CIFAR-100, and iii) experiments with different network/architecture sizes. Assuming that the reviewers proposed accepting the paper contingent on implementing the full plan for the revision, I decided to reject the paper now, but encourage the authors to make a resubmission.

(Please also note that TMLR has a single-stage review process.)

**Resubmission Of Major Revision:**

The authors may consider submitting a major revision at a later time.